# Easy-to-actuate multi-compatible truss structures with prescribed reconfiguration

Lin Ai [1], Shukun Yin [2,3], Weixia He[3], Peidong Zhang[1] & Yang Li [1,4] ✉

Multi-stable structures attract great interest because they possess special energy landscapes with domains of attraction around the stable states. Consequently, multi-stable structures have the potential to achieve prescribed reconfiguration with only a few lightweight actuators (such as shape-memory alloy springs), and do not need constant actuation to be locked at a stable state. However, most existing multi-stability designs are based on assembling bi-stable unit cells, which contain multitudes of distractive stable states, diminishing the feasibility of reconfiguration actuation. Another type is by introducing prestress together with kinematic symmetry or nonlinearity to achieve multi-stability, but the resultant structure often suffers the lack of stiffness. To help address these challenges, we firstly introduce the constraints that a truss structure is simultaneously compatible at multiple (more than two) prescribed states. Then, we solve for the design of multi-stable truss structures, named multi-compatible structures in this paper, where redundant stable states are limited. Secondly, we explore minimum energy paths connecting the designed stable states, and compute for a simple and inaccurate pulling actuation guiding the structure to transform along the computed paths. Finally, we fabricated four prototypes to demonstrate that prescribed reconfigurations with easy-actuation have been achieved and applied a quadra-stable structure to the design of a variable stiffness gripper. Altogether, our full-cycle design approach contains multi-stability design, stiffness design, minimum-energy-path finding, and pulling actuation design, which highlights the potential for designing morphing structures with lightweight actuation for practical applications.

Multi-stable morphing structures have the characteristic of being stable at more than two configurations without the need for locking by actuators. In addition, such structures have special energy landscapes where there are domains of attraction around the stable states, which might enable the structures to reconfigure with fewer inaccurate actuators. These features imply their potential to employ lightweight actuation, such as shape-memory alloy (SMA) springs. Another feature

of such structures is their possibly impulsive transformation from one shape to another by instability, like the Venus flytrap[1].

Due to the easy-to-actuate and impulsive-motion features, multi-stable morphing structures are applied in morphing air inlets[2], reconfigurable microelectronic devices[3], origami structures at the meter scale for arches or emergency shelters[4], energy harvesters[5], mechanical metamaterials for vibration isolation and energy

[1]The Institute of Technological Sciences, Wuhan University, Wuhan, Hubei 430072, China. [2]Andrew and Peggy Cherng Department of Medical Engineering, Division of Engineering and Applied Science, California Institute of Technology; Pasadena, California 91125, USA. [3]Hongyi Honor College, Wuhan University, Wuhan, Hubei 430072, China. [4]Wuhan University Shenzhen Research Institute, Shenzhen 518057, China. ✉e-mail: yang.li@whu.edu.cn

absorption[6], and robots with slow actuation but fast motion due to structural instability[7].

Most existing multi-stable structures are assembled with bistable unit cells which are based on the geometrically axial or rotational symmetry[8,9]. Although it is easy to design these units intuitively to achieve prescribed stable configurations, a large number of redundant stable states would be introduced and require complex actuation for reconfiguration[10]. One way to solve this problem is to introduce pneumatic or shape memory polymer actuation (or other forms of overall actuation) to apply forces on all bi-stable units simultaneously when overall pressured or heated, but it ends up with only two designable stable states, such as bistable auxetic surface structures consisting of bi-stable triangular auxetic units[11] and Kresling-like origami tubes with sequentially connected bi-stable elements[4]. Bi/multi-stability can also be emerged from prestressed shells, such as bistable tape springs[12], helical connection of pre-curved shell strips[13], and pre-tensioning shell frames for multiple out-of-plane buckling modes[14]. Such multi-stable compliant shell structures also have many distracting stable states and need many actuators to control[15]. A multi-DOF rigid mechanism mounted with a small number of bistable unit cells is proposed to achieve prescribed multi-stable configurations[16]. In fact, they need to manually couple and decouple some control vertices to realize multi-stability. These additional operations of coupling and decoupling again cause actuation complexity.

Another type of multi-stable structures consists of rigid mechanisms and elastic units and designs the parameters of elastic springs, and the multi-stability is achieved by designing the mechanism geometry and spring parameters (like stiffness and prestress). When the kinematic relationship is simple, a closed-form expression of all kinematic relationships can be derived consisting of the geometric constants and a single kinematic variable. As a result, the symbolically expressed elastic energy along the kinematic path can be represented and differentiated with respect to the kinematic variable to derive multi-stability constraints for the geometric constants[17,18]. Another method looks for nonmonotonically geometric variation along the kinematic path, and it puts an elastic component at that nonmonotonic feature where the elastic component goes through its rest position several times due to the nonmonotonic behavior along the kinematic path, which contributes to multi-stability[19]. However, constructing prescribed nonmonotonic behavior for a general mechanism does not seem possible. For arbitrarily complicated mechanisms with prescribed stable states, a general method by taking local expansions of nonlinear kinematic equations and extracting multi-stability constraints was proposed[20]. However, the energy barrier and stiffness of such multi-stable structures designed are generally very low.

One study has found that bi-compatibility could induce bi-stability upon kirigami tessellations restricted on a sphere[21]. In this paper, we extend this idea and propose a method for designing a type of multi-stable structure that achieves prescribed stable configurations with high local stiffness based on the proposed multi-compatibility constraints. Its comparison with other methods is provided in Fig. 1a, and Fig. 1b illustrates the graphic formation of multi-compatibility constraints, which suggests that all zero degree-of-freedom (DOF) and statically indeterminate truss structures can be geometrically redesigned into *multi-compatible structures*, i.e., multi-stable structures with elastically deformable components and multiple zero-energy states. Then, we employ a minimum-energy-path (MEP) finding method[22–26] to connect two stable states among a multi-stable energy landscape (as shown in Fig. 1c). Furthermore, a pulling actuation can be designed to follow the computed MEP (as shown in Fig. 1d).

## Results
### An inverse design method for multi-compatible truss structures
To demonstrate that the inversely designed multi-stability can be achieved by employing multi-compatibility constraints, consider designing two four-bar linkages achieving the same three prescribed coupler-link positions and orientations, as shown in Fig. 1b(I) and (II), where each prescribed position and orientation are quantified by the coordinates of $\mathbf{P}$ and the rotation angle $\theta$ (of the coupler-link to the reference geometry). By taking the first configuration as the reference, $\theta_1$ equals zero. As the solutions of four-bar linkages reaching the same prescribed positions and orientations for the coupler link are abundant, two different 4-bar linkages reaching the same prescribed coupler-link configurations are obtained as shown in Fig. 1b(I) and (II) (see Supplementary Information Section 1 for designing a four-bar linkage going through $n$ prescribed locations). By rigidly connecting the couplers of these two four-bar linkages, it forms an incorporated structure with two plates joined by three bars (one bar has been merged), as shown in Fig. 1b(III), which has zero DOF and thus immovable according to the Chebychev–Grübler–Kutzbach (C-G-K) criterion[27]. In Fig. 1b(III), when the incorporated structure passes through $[\mathbf{P}_1, \theta_1]$, $[\mathbf{P}_2, \theta_2]$, and $[\mathbf{P}_3, \theta_3]$, where the pin-jointed bars are assumed to be elastically deformable, the structure is at its compatible and thus stable states. This is because its two consisting components, i.e., the two four-bar linkages in Fig. 1b(I) and (II), can both achieve three identical coupler-link configurations, respectively. In Fig. 1b(III), it is noticed that the paths of the two four-bar linkages diverge when moving away from $[\mathbf{P}_1, \theta_1]$, $[\mathbf{P}_2, \theta_2]$, and $[\mathbf{P}_3, \theta_3]$, which indicates incompatibility. Thus, elastic deformation of pin-jointed bars is needed. There are two more path intersections, $\mathbf{U}$ and $\mathbf{N}$, in Fig. 1b(III). When considering the rotation of the coupler link, only the point $\mathbf{U}$ is an intersection, i.e. a compatible state, in the 3D plot as shown in the blue dashed box in Fig. 1b(III). As a result, we designed a quadra-stable structure where three compatible states are specified and one is redundant.

The above phenomenon is called "multi-compatibility" in this paper, and the corresponding constraints could be formulated by requesting the length of each pin-jointed bar to be equal at different prescribed configurations as shown in a dissembled dyad in Fig. 1b(II) as:

$$\|\mathbf{W}_i\| - \|\mathbf{W}_1\| = 0 \quad i = 2, \ldots, nc \tag{1}$$

$$\mathbf{Z}_i = \mathbf{Z}_1 \bullet Rot(\theta_i) \quad i = 2, \ldots, nc \tag{2}$$

where $\mathbf{W}$ and $\mathbf{Z}$ are the respective vectors representing the proximal and distal link, $\mathbf{W}_1$ denotes the proximal link at the reference configuration, while $\mathbf{W}_i$ $(i > 1)$ represents the ones at other configurations, and the same for the distal link $\mathbf{Z}$. $\theta_i$ denotes the rotation angle of the distal link $\mathbf{Z}_i$ relative to $\mathbf{Z}_1$, $nc$ denotes the number of prescribed configurations and $Rot(\bullet)$ is the rotation matrix. Moreover, the distal links $\mathbf{Z}$, which make up the plate, are constrained to be identical through the corresponding rotations as in Eq. 2 to avoid plates' flips (see Supplementary Information Section 2 for the multi-compatibility design method in detail).

Furthermore, the stability, i.e., local stiffness, at the stable states is of interest and can be designed in constrained optimization in the form of the eigenvalues of the stiffness matrix. (see Supplementary Information Section 3 for local stiffness calculation in detail). The energy barrier between stable states could also be influenced by the local stiffness design, which specifies the local curvature of the energy landscape (see Supplementary Information Section 3 for the discussion about the local stiffness in detail).

### Reconfiguration path-finding and actuation design
The above sections have proposed a design method to create a structure with several discrete stable states, but the transformation among these stable states has not been discussed, and a proper actuation to facilitate that transformation is not known. Moreover, as

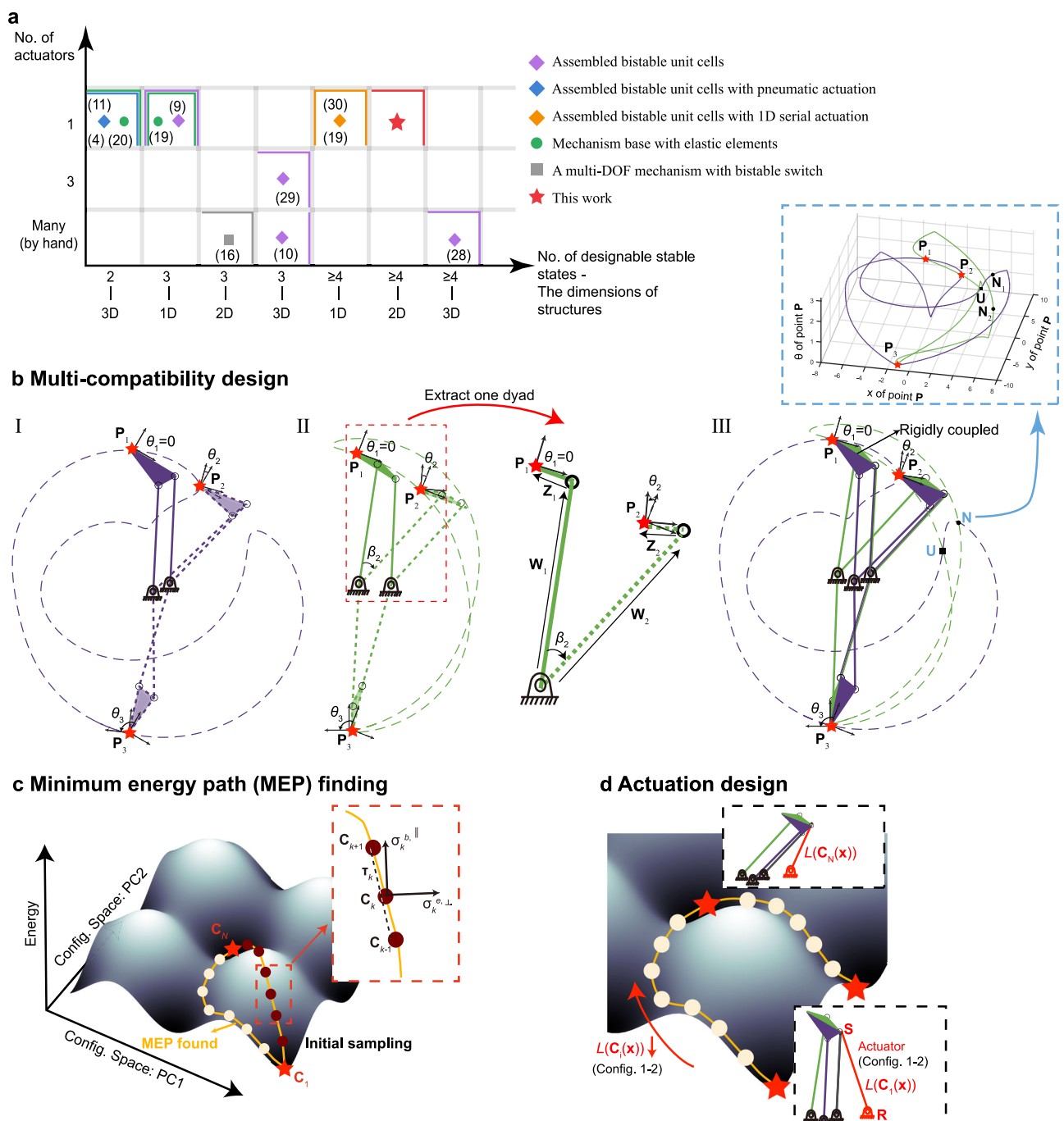

**Fig. 1 | Schematic diagram for designing a multi-stable truss structure and its actuation design. a** Literature survey of different types of the multi-stable structures in terms of their number of designable stable states and the number of actuators needed for the reconfiguration[4,9–11,16,19,20,28–30]. Heavy lines in different colors bound the upper limits of performance of the corresponding researches. **b** Illustration of the formation of multi-compatibility. (I) One four-bar linkage goes through three prescribed coupler-link configurations, denoted by the reference point **P** and its orientation $\theta$. (II) Another four-bar linkage goes through the same prescribed coupler-link configurations as (I). A dissembled dyad with geometric and kinematic parameters is shown on the right. (III) Rigidly connect the couplers of these two four-bar linkages, then the union becomes an immovable structure and is multi-compatible at three prescribed coupler-link configurations $P_1$, $P_2$, $P_3$, represented by the intersections of the trajectories of two four-bar linkages in (I)

and (II). There are two more path intersections in the 2D plot, **U** and **N**. When considering the rotation of the coupler link to plot trajectories in 3D space in the blue dashed box, only the point **U** is an intersection, i.e. a compatible state. **c** Illustration of minimum-energy-path (MEP) finding method connecting specified stable states by regarding the bars as elastic translational springs and employing Nudged Elastic Band (NEB) method, where initially linearly sampled configurable points iteratively move towards the valleys of the energy landscape. **d** Illustration of one actuation design method by finding two nodes (**R** and **S**) on the structure where the distance L in between decreases monotonically along the computed MEP, and consequently, a simple actuation by shortening the distance between these two nodes can be applied so that the structure might reconfigure along the computed path and reach the desired stable states without being bifurcated and falling into wrong stable states.

the complexity of the structure increases, i.e., with a larger number of components, more redundant stable states exist (see Supplementary Information Section 11.3), which can cause transformation bifurcation. As a result, we need to explore an easy-to-achieve path, i.e., minimum-energy-path (MEP), connecting the target configurations. Therefore, we employ the nudged elastic band (NEB) method[26], which is widely used in computational chemistry and introduced to the nonlinear structural mechanics to calculate MEP in recent years[22–25], to explore an MEP connecting two stable states. Figure 1c shows the principle of the NEB method in the configuration space after the dimensionality reduction by the Principal Component Analysis (PCA) method, where $\mathbf{C}_1$ and $\mathbf{C}_N$ represent the configuration points of the two respective stable states that are designed (see Supplementary Information Section 4.2 for the NEB method in detail.) The orange line in Fig. 1c shows the schematic diagram of the MEP found by the NEB method.

Then, we design a simple actuation, where only pulling is allowed in order to employ string or smart material (like shape memory alloy springs) lightweight actuators to guide the structure to morph along the MEP. The positions of endpoints of the pulling actuator at the reference configuration (in Fig. 1d) are the design variables, where the ground endpoint is $\mathbf{R} = (x_g, y_g)$ and the moving endpoint is $\mathbf{S} = (x_a, y_a)$ as shown in Fig. 1d. The length of the actuator can be thus represented by $\mathbf{x} = [x_g, y_g, x_a, y_a]$ as $L(\mathbf{C}_k(\mathbf{x}))$ along the computed MEP. The constraint for "only pulling allowed actuation" is that $L(\mathbf{C}_k(\mathbf{x}))$ varies monotonically along $k = 1, ..., N$, as shown in Fig. 1d. The cost function is to maximize the variation of the actuator length $|L(\mathbf{C}_1(\mathbf{x})) - L(\mathbf{C}_N(\mathbf{x}))|$, which is to minimize the required pulling force, and more inequality constraints could be included to restrain the mounting locations. Therefore, the actuation design problem could also be formulated as a constrained optimization problem:

$$
\begin{aligned}
&\text{For } \mathbf{x} = [x_a, y_a, x_g, x_g]: \\
&\min -\left|L(\mathbf{C}_1(\mathbf{x})) - L(\mathbf{C}_N(\mathbf{x}))\right| \\
&\text{subject to } L(\mathbf{C}_k(\mathbf{x})) > L(\mathbf{C}_{k+1}(\mathbf{x})) \text{ for } \mathrm{k} = 1, ...,(N-1) \\
&lb < x_a, y_a, x_g, x_g < ub
\end{aligned}
\tag{3}
$$

The design procedure involves a three-step optimization: (1) compute the multi-compatible structure geometry and stiffness; (2) calculate the MEP by NEB method; (3) design the corresponding actuation. A higher-level loop has been applied on top of the three-step optimization for a more compact actuation design and energy barrier control. The detailed algorithms are provided in Supplementary Information Section 5. The three optimization problems can be merged into one in the future for better computational efficiency.

## Examples: multi-compatibility design, MEP finding, and pulling actuation design

In this paper, we designed the multi-compatible structures and actuators in sequence. Consider designing a two-plate-four-bar (2P4B) structure with three prescribed stable configurations as shown in Fig. 2a. The grey plate is fixed at the ground, and the other plate can reach three prescribed configurations in three colors, denoted by the reference point $\mathbf{P}$ and its orientation angle as listed in Supplementary Information Section 11.1. (See Supplementary Information Section 11.4 for the step-by-step design procedure of the 2P4B structure).

We solve the optimization problem numerically using MATLAB's built-in optimization routine "fmincon" (see Supplementary Code 1 for the 2P4B structure design, Supplementary Code 2 for the design toolbox for customizing the multi-stable plate-and-bar structure.) The corresponding physical model is shown in Fig. 2b. The components were designed in SolidWorks and 3D printed using Ultimaker S5, and they were then connected by out-of-shelf revolute joints. To avoid self-contact, components were assigned to different layers. However, some pillars protruding from the joints would hinder the rotation of some

bars, so the plate employs bypassing designs, as shown in Fig. 2b, where a cantilever component is attached to the moving plate to bypass the moving trajectory of bars. Another method to avoid self-contact refers to the design of a crankshaft which has a higher strength requirement. Thus, we adopted the cantilever-bypassing-plate design. As the structure shall suffer large deformation during the reconfiguration, two white bars that are made of a rubber-like material, TPU, are employed, and the transformation process is given in Fig. 2b. It shows that the two white bars deform significantly and bounce back to the original shape when reaching the three designed stable states. (See Supplementary Movie 1 for its reconfiguration).

To demonstrate the generality of this method, a more complex example, a three-plate-six-bar (3P6B) structure, is carried out as shown in Fig. 2c. The grey plate is fixed to the ground, and the other two plates both can achieve three prescribed configurations respectively as listed in Supplementary Information Section 11.1. Figure 2d presents the corresponding physical model, and the structure is multi-compatible at three prescribed configurations while bars deform significantly during the transformation (see Supplementary Movie 2 for its reconfiguration). As the complexity of the structure increases, the components interfere more seriously, and a dedicated algorithm for assigning the components into different layers is developed (see Supplementary Information Section 6.2 and 6.3 for the algorithm of the component assignment). Another tri-stable structure in 3D employing universal joints has also been designed. Because the interference of components is unavoidable in 3D, the physical model is not manufactured, and only simulation is conducted in Supplementary Fig. 16 (see Supplementary Information Section 7 and Section 10.1 for a 3D tri-stable example.) Furthermore, the inverse design problem is a nonlinearly constrained optimization problem, so the condition of solutions is hard to predict. Some parametric studies are conducted to explore how design parameters would influence the design space and their properties, for instance, the maximum number of stable states that we can design, the number of total stable states including global and local minimums, and the value of local stiffness, etc. (See Supplementary Information Section 10.2 for parametric studies of the design method).

The above examples illustrate the feasibility of multi-compatibility design, but they are cumbersome to manufacture due to the component layering. We attempted to propose the interference-free constraints (see Supplementary Information Section 6.1) so that we can design an interference-free 2D two-plate-three-bar example as shown in Fig. 2e, where all components could be arranged in one plane after reshaping two plates. The grey plate is fixed to the ground, and the targets of the moving plate are prescribed as listed in Supplementary Information Section 11.1. The physical model of this unit is shown in Fig. 2f, where the three designed stable states are verified. Then, we sequentially joined four of such units to form a multi-stable morphing structure in Fig. 2g, where the overall curvature changes (see Supplementary Movie 3 for its reconfiguration.) Such easy-to-manufacture multi-compatible unit cells might serve as building blocks for more complex morphing structures and mechanical metamaterials.

Moreover, to illustrate the compatibility design with the local stiffness design, another two-plate-three-bar quadra-stable structure with different local stiffness at different configurations is designed in Fig. 3. In addition to the prescribed configurations listed in Supplementary Information Section 11.1, the constraints about the local stiffness are added as follows:

$$
k_{c1}, k_{c4} > v_1, \; k_{c2}, k_{c3} < v_2
\tag{4}
$$

where $k_{ci}$ (i = 1, 2, 3, 4) denotes the local stiffness at the $i$th configuration. The first configuration and the last configuration are expected to be locked tightly with higher stiffness values that are higher than $v_1$, while the middle configurations are expected to transform easily with

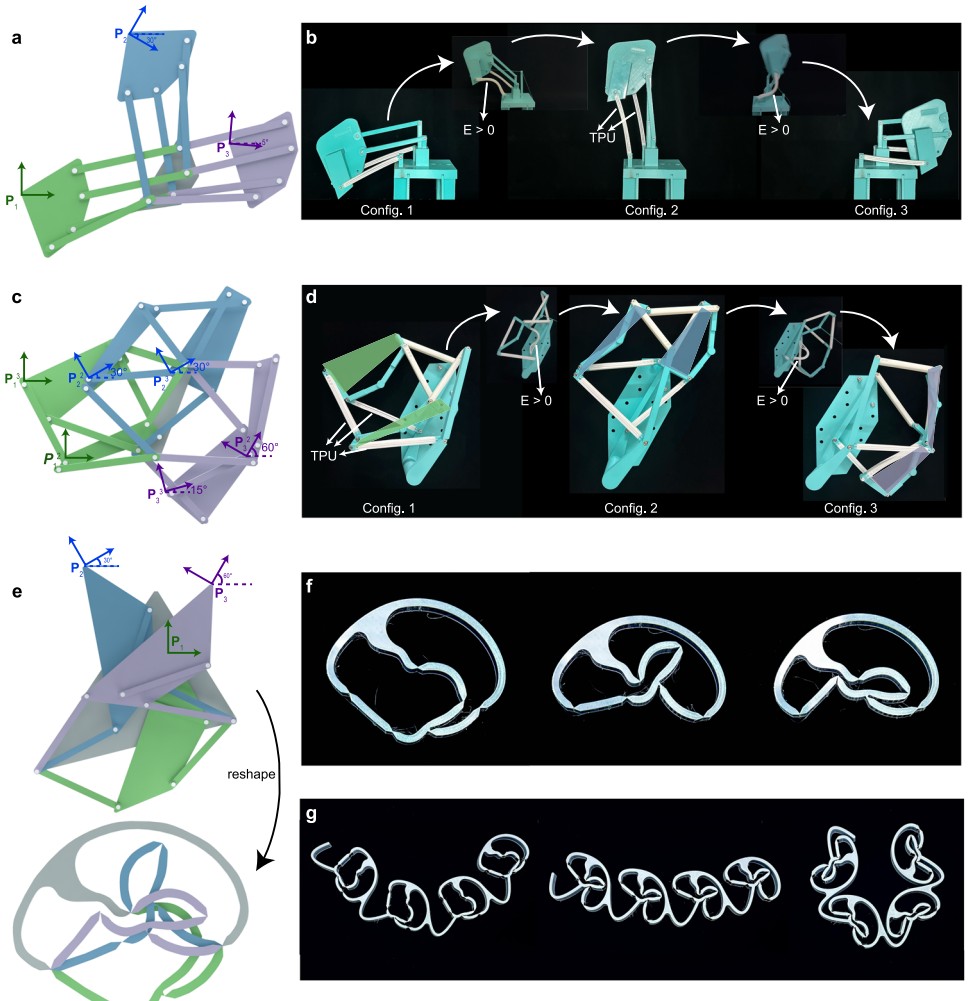

**Fig. 2 | Examples of the multi-compatibility design. a** A two-plate-four-bar tristable structure (2P4B). The grey plate is fixed to the ground, the moving plate can achieve three prescribed configurations, denoted by three respective colors (green, blue, and purple). **b** The physical model corresponding to **a**, where the structure is multi-compatible at three prescribed configurations without component deformation, while the white bars made of TPU deform significantly during the reconfiguration. **c** A three-plate-six-bar tri-stable structure (3P6B). The grey plate is fixed to the ground, the other two moving plates can achieve three prescribed configurations, denoted by three colors (green, blue, and purple). **d** The physical model corresponding to **c** where the components show significant deformation during reconfiguration. **e** A design of a real-2D two-plate-three-bar tri-stable structure, where all components could be arranged in one plane after reshaping, is obtained by adding the interference-free constraints. **f** The physical model corresponding to **e**. **g** A sequential assembly of four units in **f**, which can be stable with three different curvatures.

lower stiffness values that are lower than $v_2$. Their optimized local stiffnesses are listed on the energy curve of the designed quadra-stable structure, and it achieves four prescribed compatible configurations as shown in Fig. 3a. The physical model at the prescribed compatible states is shown as the first row of figures in Fig. 3b, and three subfigures in orange boxes demonstrate the configurations with the highest strain energy during the reconfiguration (see Supplementary Movie 4 for its reconfiguration actuated by only one rotational handle). Then, assemble three such unit cells, as shown in Fig. 3c, where the reference points of each unit cell, which are moved a little to avoid interference in the physical model, are marked in yellow. We can observe the outline of the reference points is shrinking, so the assembling multi-stable structure could be a gripper with multiple stiffnesses if some attachments are added to the reference point of each unit.

Then, we conducted the MEP finding and actuation design for the two-plate-four-bar (2P4B) and three-plate-six-bar (3P6B) examples in Fig. 2a and c, respectively. We applied the truss model, i.e. each bar is a simple two-node linear member that only takes axial extension or compression, in the MEP finding for high efficiency

(see Supplementary Information Section 4.1 for the discussions on the truss model and beam model in detail.) For the 2P4B example, the stiffness of the plate element (i.e., truss after triangulation) is set to 10 (N/m), the stiffness of the bar element is 1 (N/m), and the stiffness of the elastic bands is 10 (N/m). Figure 4a demonstrates the MEP of the 2P4B structure under PCA projection (see Supplementary Information Section 11.2 for discussions on the transformation between the relevant degrees of freedom), and the black curve represents the path between the Config. 1 and Config. 2, while the red curve is the path between Config. 2 and Config. 3. A series of configurations along the path is presented in Fig. 4e, and the corresponding pulling actuator is noted. In Fig. 4b, the path is plotted together with the equipotential surface of energy in the configuration space (without the dimensionality reduction) in terms of the coordinates $[\mathbf{P}_x, \mathbf{P}_y, \theta]$ of the reference point in the moving plate. The path is shown to be wrapped within the low-energy equipotential surface where the endpoints of the path converge to zero energy. This confirms the effectiveness of the NEB method in finding MEP. For the 3P6B example, the stiffness of the triangulated bars that make up the plate is set to 40 (N/m), the stiffness

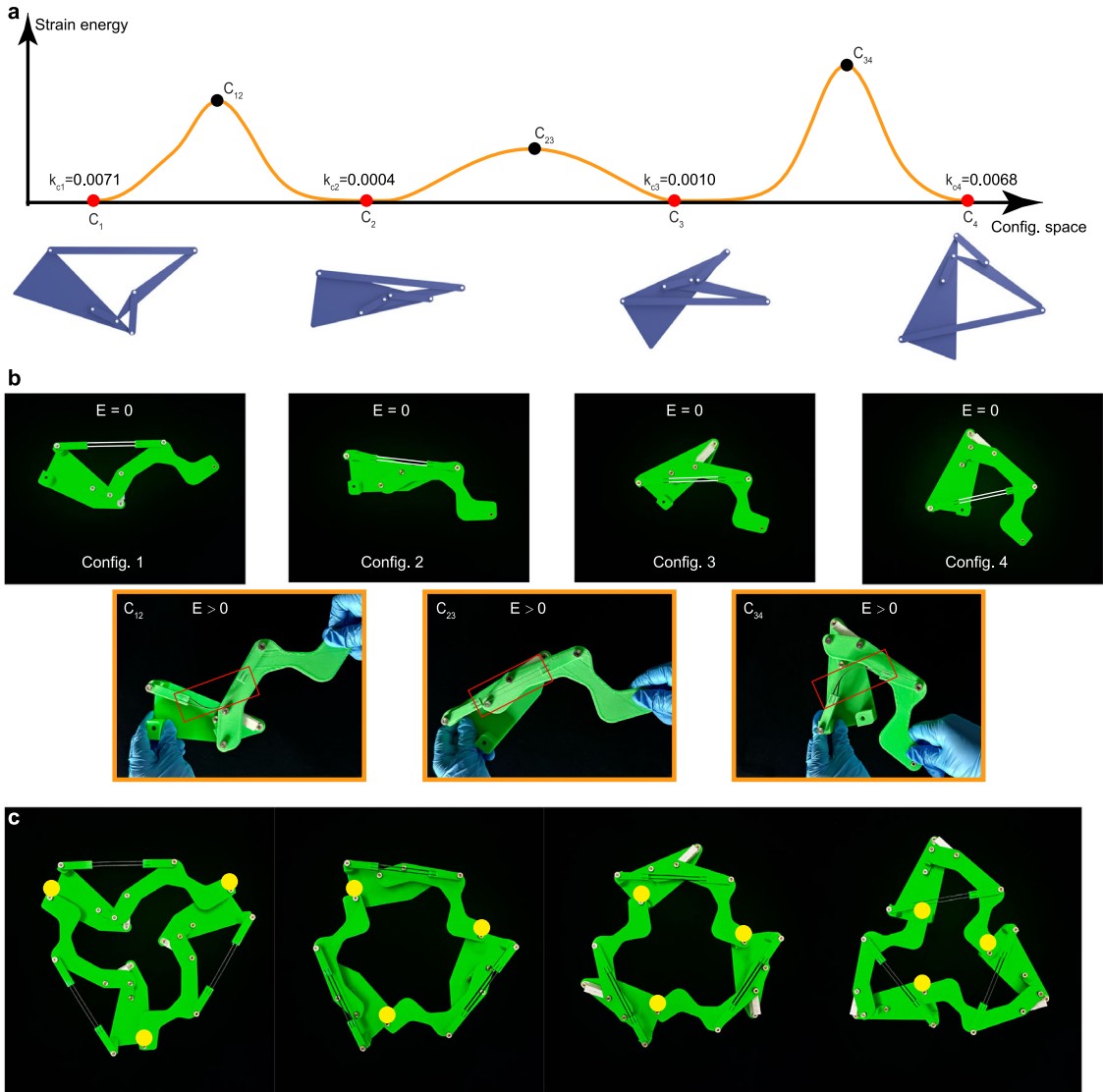

**Fig. 3 | The results of local stiffness design example, a two-plate-three-bar quadra-stable structure. a** The energy curve of the designed structure and four designed compatible configurations. The local stiffness of the first and last configurations is designed to be larger for locking tightly, while the local stiffness of the middle configurations is to be smaller for transforming easily. The energy barrier is plotted as the orange curve, where the energy barrier of the transformation between two configurations with lower stiffness is obviously lower than others. **b** The physical model of this quadra-stable structure. The first row of figures is four prescribed stable states corresponding to the red points in **a**, where two thin white lines denote two slices of spring steel. Three subfigures in the orange boxes show the configurations with the highest strain energy (characterized in the red boxes). **c** Use the multi-stable unit cells in **b** to construct a multi-stable structure whose outline of the reference points is shrinking. This multi-stable structure could be a reconfigurable and variable-stiffness gripper when adding attachments at reference points.

of the bars interconnecting the plates is 0.1 (N/m), and the stiffness of the elastic bands is 0.5 (N/m). Figure 4c demonstrates the MEP of the 3P6B example under PCA projection, where two additional locally stable states are observed, and their corresponding physical models are presented in Supplementary Fig. 21. A series of configurations along the MEP is presented in Fig. 4f, with the corresponding pulling actuator being marked as yellow and orange bars. Figure 4d presents the dimension-reduced energy landscape of the 3P6B example, in which we set the z-axis as $ln(E_e + 0.005)$, aiming to expand the lower energy region, and the path in Fig. 4c could be observed clearly in Fig. 4d (sketched by a red curve). Furthermore, the physical model of the 3P6B example has at least five additional stable states besides the prescribed three stable configurations (see Supplementary Information, Supplementary Fig. 21, in Section 11.3 for the photographs of the five additional stable states.) These additional stable states could also be predicted by Fig. 4d, as there are a myriad of low-energy states

(shown in dark blue) around the MEP path. In experiments, if we pull the physical model intuitively, it would be easily trapped in an unexpected stable state (see Supplementary Movie 2.) Additionally, the curvature of the MEP reflects the stability of each configuration. As a result, we find that computing MEP and correspondingly designing the actuation are crucial in the successful navigation of the desired reconfiguration when many diverging stable states are present (see Supplementary Information Section 10.3 for applying the MEP finding and actuation design on a more complex structure assembled by two quadra-stable unit cells).

Figure 4e and f demonstrate the computed pulling actuation locations of the respective 2P4B and 3P6B structures (see Supplementary Movie 1 and 2 for the reconfiguration experiments that employ the designed pulling actuators.) We can observe the length of the actuator decreases monotonically during the transformation. We performed experiments employing the designed actuation, and the

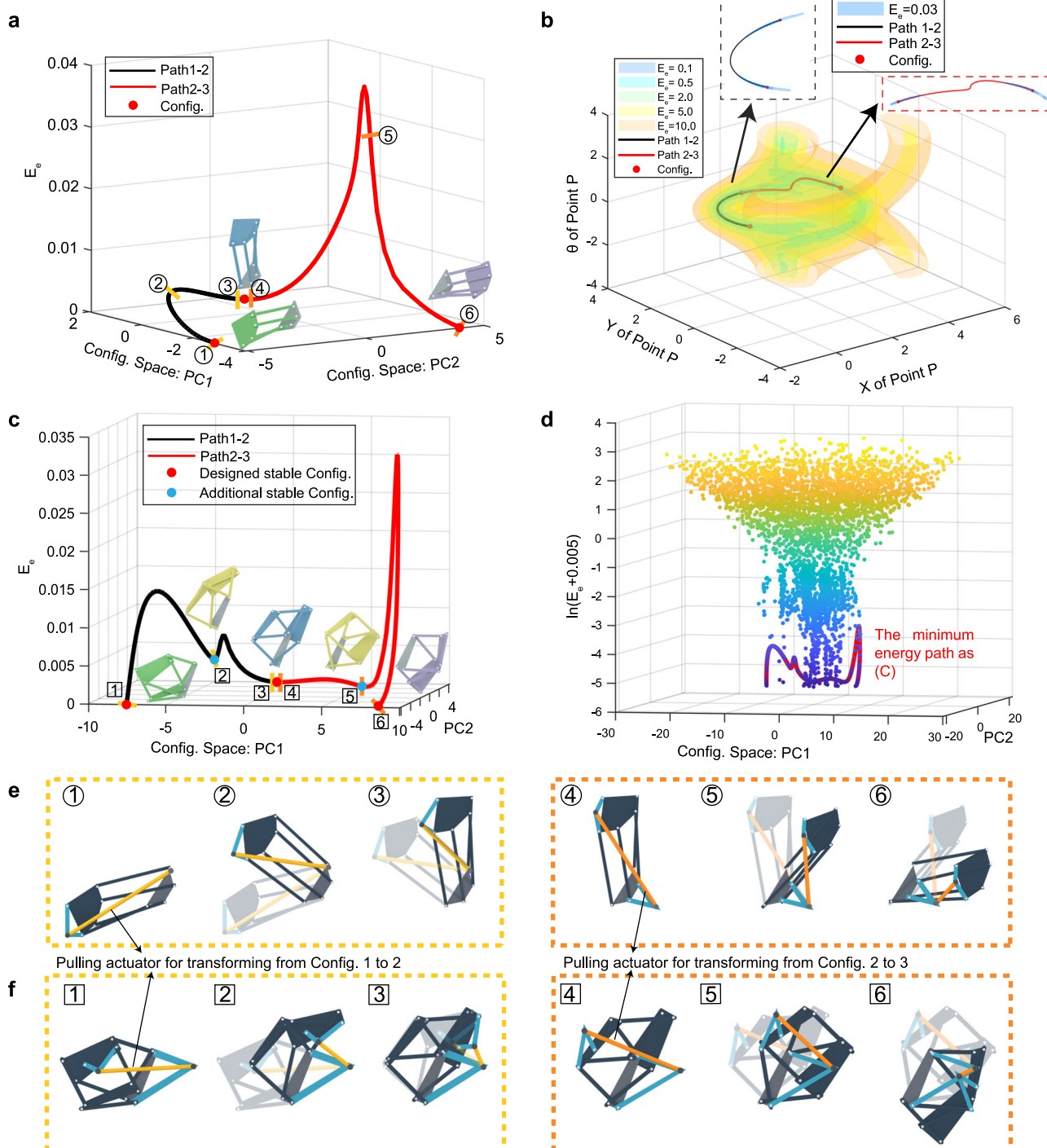

**Fig. 4 | Results of the MEP finding and the pulling actuation design. a** The computed MEP of the 2P4B structure acquired from the NEB method is plotted, where the axises are respectively the first and second principal components of the configuration space and the corresponding strain energy $E_e$. Red dots represent the three designed compatible configurations. Numbers in circles correspond to the intermediate configurations shown in **e. b** The MEP is reparametrized and plotted in the coordinating space $[x_p, y_p, \theta]$ of the reference point **P** in the moving plate. The path is in the low-energy region, wrapped by energy equipotential surfaces. **c** The MEP of the 3P6B structure found by the NEB method. Red dots

represent the three designed stable configurations which are shown beside the curve in green, blue and purple geometries respectively. Blue dots on the path represent the additional stable configurations that are shown in yellow beside the curve. Numbers in boxes on the path correspond to the intermediate configurations shown in **f. d** The energy landscape of the 3P6B with the MEP being highlighted with a red curve. A logarithm transformation of the strain energy $ln(E_e + 0.005)$ is applied for the z-axis. **e, f** The computed intermediate configurations with the pulling actuation. Actuators are represented by the yellow and orange lines. Actuator mounting locations are supported by the light blue dyads.

setup is presented in Supplementary Fig. 15a where the cable forces and their contraction displacements can be recorded. In Figs. 5 and 6, the recorded force-displacement curves of these two structures and the corresponding snapshots are presented. The recorded curve is not

smooth due to the measurement fluctuation introduced by the time-dependent relaxation of TPU. In the curves shown in Fig. 5, the force in the cable decreases to zero when the structure starts to morph to a stable state automatically and releases strain energy, i.e., captured by

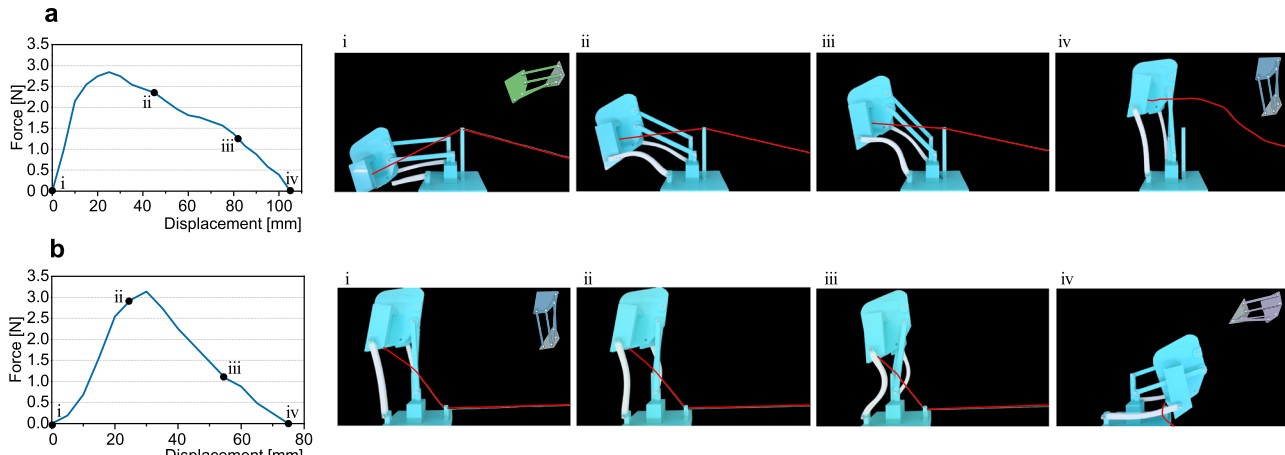

**Fig. 5 | Actuation experiments of the two-plate-four-bar (2P4B) structure and the corresponding force-displacement curves. a** The force-displacement curve of the 2P4B structure from Config.1 to Config.2. The force decreases to zero, indicating that the structure reaches a stable state. **b** The force-displacement curve of the 2P4B structure from Config.2 to Config.3.

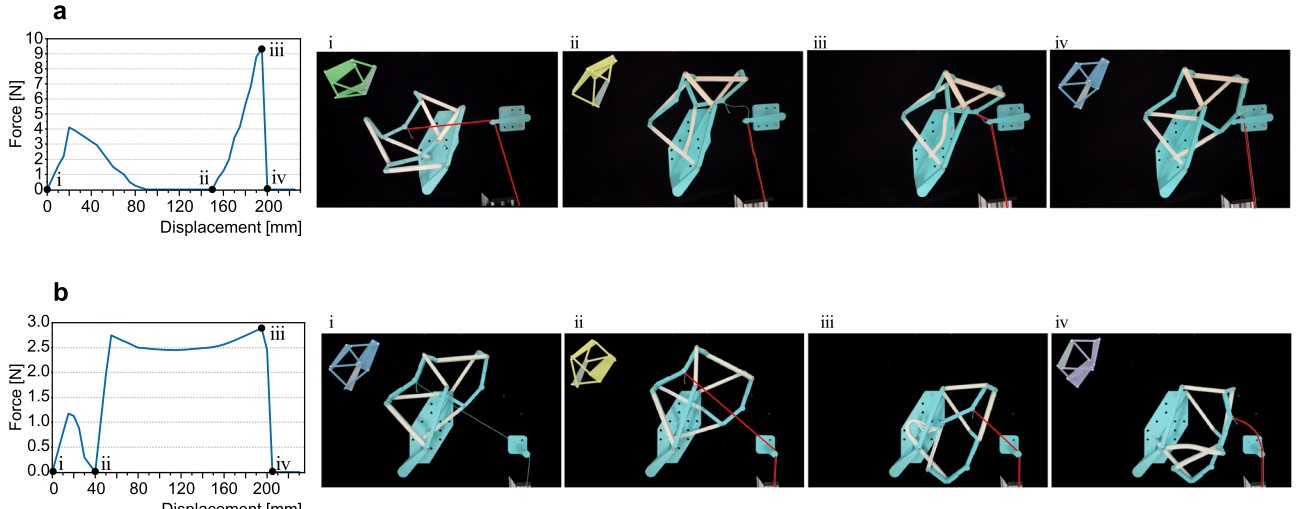

**Fig. 6 | Actuation experiments of the three-plate-six-bar (3P6B) structure and the corresponding force-displacement curves. The additional stable configurations predicted by the NEB method in Fig. 3c are noted in a(ii) and b(ii). a** The actuation experiment of the 3P6B structure from Config. 1 to Config. 2. In the force-displacement curve, the force decreases to zero in the middle, indicating that an additional stable state exists, which is the same as in Fig. 3c. **b** The actuation experiment from Config. 2 to Config. 3.

the domain of attraction. The two intermediate stable states of the 3P6B structure predicted in Fig. 4c are verified by the experiments in Fig. 6a(ii) and b(ii). Furthermore, the actuation cable can be fabricated by the temperature or magnetic field-sensitive raw materials in the future so that the structure can automatically reconfigure under different external fields.

### The application demonstration case of a quadra-stable structure for a variable stiffness gripper

To explore the possible application of the proposed easy-to-actuate multi-stable structures, a variable stiffness gripper is designed employing a quadra-stable structure (Fig. 7a). It can reconfigure between 4 stable states with only 1 actuator. The reconfiguration between 2 (out of 4) stable states gives the stiff gripping mode, while the other 2 stable states provide the soft mode. One actuator is enough to operate the gripper in the two respective modes as well as switch between the two modes. Figure 7b presents the specific force data of the gripper, where the gripping force in the stiff mode is around 30

times of the force in the soft mode. The measurement setup is presented in Supplementary Information Section 9. To demonstrate the different performance of the two modes, the gripper crushing the tofu in the stiff mode while gently picking it up in the soft mode is presented in Fig. 7c. The whole process is facilitated by only one actuator as shown in Supplementary Movie 5. The comparison of this quadra-stable gripper with the bi/multi-stable grippers in the literature is provided in Fig. 7d, where the gripper in this paper is the only one that can achieve bistability and variable stiffness with 1 actuator. This proves the effectiveness of our method in designing easy-to-actuate multi-stable structures. In the future, the shape and compliance of the end effectors can be designed carefully for better object holding, and the scaling-down for medical endoscope devices can be carried out.

### Discussion

In this paper, we have proposed a method of designing multi-compatible truss structures with (more than two) target

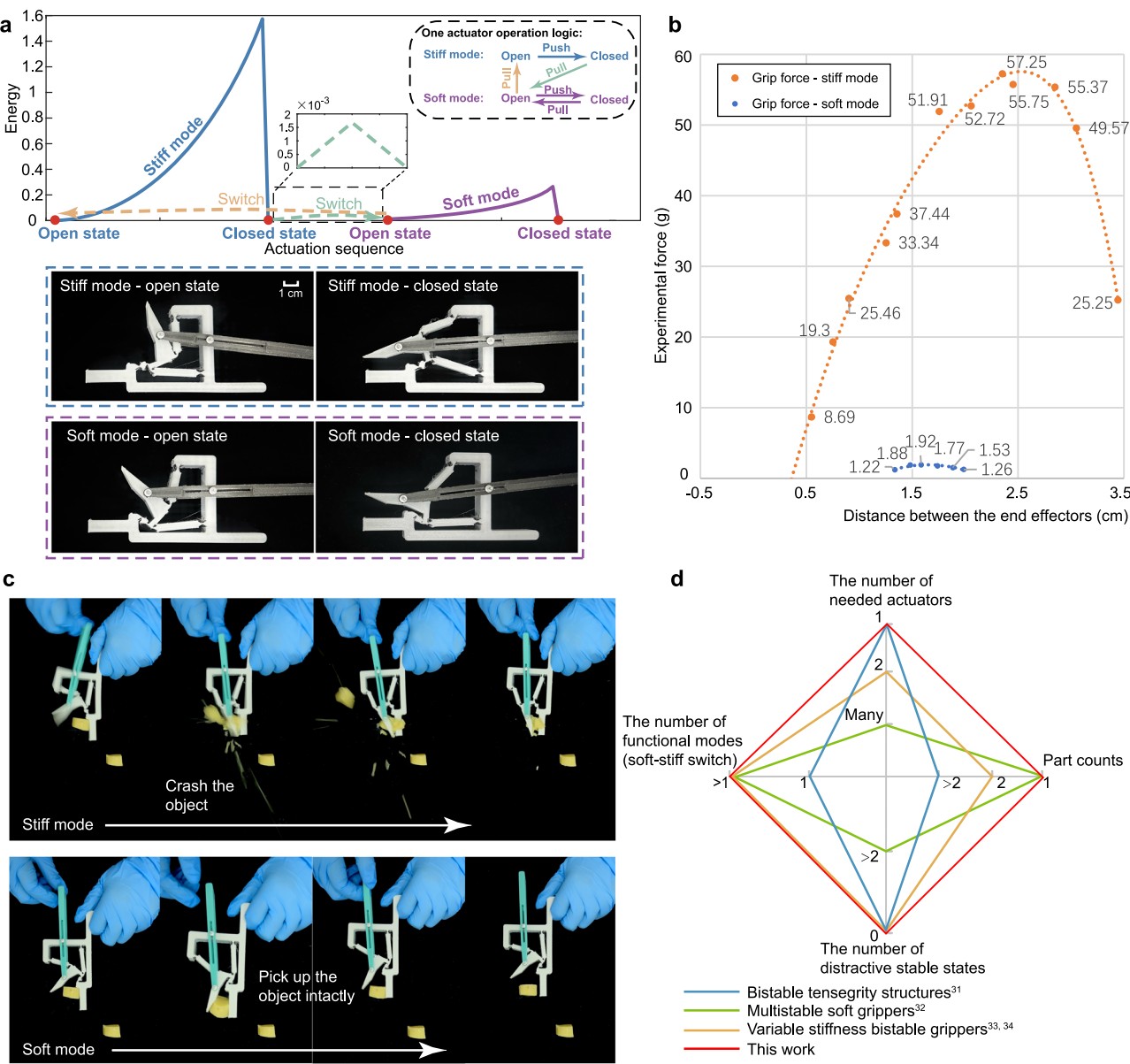

**Fig. 7 | Variable stiffness quadra-stable gripper. a** Demonstration of a gripper with both stiff mode and soft mode based on a quadra-stable structure. Two of stable states are the open and closed state of the gripper in stiff mode, where the energy barrier during the reconfiguration is much higher than that between the other two of stable states in soft mode. One actuator is enough to operate the gripper in the two respective modes as well as switch between the two modes. **b** The grip force data of the gripper by the experiment. With different opening distances between the gripper's end effectors, the variations of the gripping force, i.e. a constant force, in the stiff mode and soft mode are denoted by the orange curve and the blue curve. The gripping force in the stiff mode is around 30 times of the force in the soft mode. **c** The performance that the gripper grabbed a tofu in different modes. The gripper crushed the tofu in the stiff mode while picked up the tofu intactly in the soft mode (see Supplementary Movie 5 for the whole process.) **d** Different types of bi/multi-stable grippers in the literature that are compared in terms of the number of needed actuators, the number of functional modes, the number of distractive stable states and part counts[31–34].

configurations specified by a series of reference points. The design can be framed as a constrained optimization problem employing "multi-compatibility" constraints. The stability at stable states, i.e., the local stiffness, could be incorporated into the optimization and thus prescribed. Then, we applied the nudged elastic band method to find the minimum energy path connecting the specified stable states and correspondingly designed an inaccurate pulling actuator guiding the structure to reconfigure along the computed MEP path. Some easy-to-manufacture multi-stable unit cells are introduced, which can be the building blocks and assembled into more versatile morphing structures and metamaterials. A reconfigurable gripper actuated by the single actuator to work in both stiff and soft modes is designed based on a quadra-stable structure. As a result, our approach leads to easy-to-actuate morphing structures with (more than two) specified target shapes, which might allow us to employ only a small number of lightweight and inaccurate actuators such as SMA springs and also have designable stiffness. In the future, this method could be applied for practical and lightweight applications such as morphing wings and mechanically reconfigurable antennas.

## Methods

The details of designing four-bar linkages and multi-stable truss structures with prescribed reconfiguration are demonstrated in Supplementary Information Sections 1 and 2. The details of the local

stiffness characterization are illustrated in Supplementary Information Section 3. The NEB algorithm is described in Supplementary Information Section 4. The strategies of collision avoidance are summarized in Supplementary Information Section 6. The explicit dynamic simulation method is described in Supplementary Information Section 7.

PLA (Polylactic acid), TPU (Thermoplastic polyurethanes) and CFRP (Carbon fiber reinforced polymer) filaments from Polymaker (China) were used for 3D printing, and Ultimaker S-series 3D printers are used for all 3D printing tasks. The fabrication process in detail is illustrated in Supplementary Information Section 8. The force measuring experiment setup is described in Supplementary Information Section 9.

## Data availability
All the data supporting the conclusions of this study are included in the article and the Supplementary Information. Source data are provided with this paper.

## Code availability
Two versions of MATLAB codes are included. Supplementary Code 1 presents the step-by-step design procedure for the simplest example (the 2P4B structure), and Supplementary Code 2 provides a general design toolbox for customizing the multi-stable structure. More codes are available from the corresponding author upon reasonable request.

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

## Acknowledgements

This work was supported by the National Natural Science Foundation of China (Grant No. 12202320, awarded to Y.L.) and the Guangdong Basic and Applied Basic Research Foundation (Grant No. 2021A1515110589, awarded to Y.L.). The authors would like to thank Wuhan University Student Engineering Training and Innovation Practice Center for technical support in manufacturing the prototypes. We would like to

acknowledge Dr. Jingyi Yang for her help in generating the idea of tailoring the energy barrier. We would like to thank Dr. Hao Zhou for his help in discussing the algorithm for merging three design steps. We would like to thank Zihan Zhou and Wenbin Yi for their help in measuring the force data of the gripper.

## Author contributions

L.A.: performed research, analyzed data, performed the experimental studies, wrote original draft, revised the paper; S.Y.: performed research, parametric study, wrote original draft; W.H.: manufactured prototypes, performed the experimental studies; P.Z.: performed preliminary study on the NEB method, improved the prototype demonstration; Y.L.: proposed the conceptualization, performed research, supervised the work, reviewed and edited the paper.

## Competing interests

The authors declare no competing interests.
