## [Peer Review File · Nature Communications]

Easy-to-actuate multi-compatible truss structures with prescribed reconfigurationREVIEWER COMMENTS

Reviewer #1 (Remarks to the Author):

The submitted manuscript titled "Easy-to-actuate multi-compatible truss structures with prescribed reconfiguration" presents a novel method to synthesize multistable mechanisms that eliminate the redundant stable configurations to a large extent. Moreover, the manuscript develops a method to design simple approximate actuation by determining the minimum energy path using the nudged elastic band method. To my knowledge, generating precise, stable configurations using multi-compatibility constraints is novel and provides a new way of developing lightweight multistable mechanisms. The manuscript further explains how MEPs can be used to design efficient inaccurate actuators. The manuscripts say, "NEB widely used in computational chemistry to explore MEP." In the past five years, some manuscripts have introduced MEP to solve MEP problems in nonlinear structural mechanics. The authors should clearly state that. However, using potential landscape exploration algorithms to obtain an approximate but effective actuation is of great value for designers seeking to develop actuation systems for complex nonlinear mechanisms. The manuscript is well written, and ideas have been communicated well. Further, the manuscript provides sufficient evidence supporting the claims made in the conclusion.

While overall, the manuscript almost meets the expected field standard, some revisions are suggested to improve the manuscript's readability and ensure that all the results shown in the manuscript are reproducible. The suggested revisions are listed below:

1. Page 1 and 2: The authors have made certain general remarks that most multistable literature structures are assembled bistable unit cells. This remark is not valid, as there are several examples of multistable structures that do not constitute bistable repeating cells:
 - a. Risso G, Ermanni P. Multi-stability of fiber reinforced polymer frames with different geometries. *Composite Structures*. 2023 Jun 1;313:116958.
 - b. Pirrera A, Lachenal X, Daynes S, Weaver PM, Chenchiah IV. Multistable cylindrical lattices. *Journal of the Mechanics and Physics of Solids*. 2013 Nov 1;61(11):2087-107.
2. As pointed out earlier, the literature on structural mechanics has shown MEP using methods like nudged elastic band, dimer, and conjugate peak refinement. Further, it is possible to tailor the energy barriers with alternate methods of producing multistable structures.
3. The work in the manuscript is at the intersection of robotics (rigid-body mechanics) and continuum mechanics. Hence, it is essential to provide references to seemingly prominent concepts like the Chebychev-Grubler-Kutzback (CGK) criterion for better readability.
4. Fig. 1 explains how to build multi-compatible truss structures. In Fig. 1B, please explicitly show that the incorporated structure has a common proximal link. Moreover, the manuscript identifies two additional intersection points, U and N (which point to additional stable states apart from the three target equilibrium configurations). What makes the intersection point U the only compatible point needs to be clarified in the text.
5. Check Eq. 5 and 6. There could be an issue with the Microsoft Word version. Eq. 6 needs to be printed correctly.
6. From the initial description of Fig. 1C, it needs to be clarified what Config Space: PC1 and Config Space PC2 mean. Please clarify the axes in the text.
7. Page 10 states that the MEP is plotted under Principal Component Analysis (PCA) projection. This line explains Point 6 to an extent. However, the details of the transformation between the relevant degrees of freedom should be included in the Supplementary information document.

Comments regarding the supplementary information document:

Overall, the supplementary information document is also well compiled. Considering the manuscript's multi-disciplinary nature, the authors should include a step-by-step detailed construction of at least the 2P4B with three prescribed conditions. It is strongly recommended that the authors include the working code of one example as a part of the supplementary material. It is strongly recommended that the PCA be applied to one of the examples to construct the MEP plots, as shown.

Reviewer #2 (Remarks to the Author):

In recent years, multi-stable structures have attracted the interest of scientists and engineers due to their special variable configurations and tunable performance. In their work, the authors proposed a method by a series of reference points to design multi-stable structures with numerous configurations. More importantly, the nudged elastic band method has been employed to understand the fundamental principle of the minimum energy path connecting the specified stable states. After that, some multi-stable structures were designed and tested to validate the effectiveness of the proposed method.

From my point of view, the work presented in this paper is interesting and this topic deserves to be studied. The following points should be further considered before publication.

1. The authors declared in the abstract that bi-stable unit cells contain multitudes of additional and distractive stable states (lines 16-18). I agree with this statement. However, the examples of multi-stable structures given in this work also include some units that will have unstable behaviors during the process of configuration change. Is the direction of instability of these units controllable, or is it random?
2. In line 99 shows that the pin-jointed bars in the model are deformable. Are their elastic deformations controllable or quantifiable?
3. The samples shown in Fig.2 F and G were fabricated by TPU and carbon fiber. Is the interface between TPU and carbon fiber well connected? The authors declare that the function of the carbon fiber is improving the stiffness. Will the stiffness of the bars affect the multi-stability of the structure?

Minor issues:

4. The full name of the technical terms should be given before using the abbreviation (line 15).
5. "our-of-shelf" in section 7 in Supplementary material should be "out-of-shelf".
6. References should be in a unified format. For example, the publication of reference 14 and the author information of reference 17 is missing.

open question:

7. Is it possible to use temperature or magnetic field-sensitive raw materials to fabricate the proposed structures so that they can automatically change configurations under different temperature or magnetic fields?

Reviewer #3 (Remarks to the Author):

The paper appears to be technically sound, proposing a problem, developing a solution, and demonstrating it.

The motivation of the work could be stronger. Background is given on lines 37-41 about the applications of other multi-stable structures, though it is unclear how this work fits within that context. Could it be applied generally to any of those applications, or is there a limited set to which the work applies? If this work does apply to any of those applications, what are the tradeoffs between designs? The paper would be strengthened by more there. Related, on lines 230-231, the authors mention that this could be used for a gripper with varying stiffness. On line 299, the authors mention it could be used for morphing wings or mechanically reconfigurable antennas. These are mentioned but it is not clear if or how it could be applied. What are the tradeoffs between this design and the work of others? The work would be significantly more compelling if there were a significant use case for the work.

In Figure 1, there is a red star to represent the authors' work. Highlighting or using some method to make it more clear where this work fits within the context of others would be helpful.

In Figure 1 on the x axis, there is "1D", "2D", or "3D" below each of the designable states. It is unclear what this refers to.

Reviewer #4 (Remarks to the Author):

Multi-stable structures are especially attractive for the application of morphing because they can be switched from one stable configuration to another with little energy and can be locked in any of the stable configurations without the need for constant actuation. However, most multi-stable structures consist of bistable unit cells and have undesirable stable configurations, which pose challenges in transforming to required states using easy-to-implement actuation techniques. The authors proposed multi-compatible truss structures with limited redundant stable states, obtained the minimum energy paths for switching between required states, and demonstrated the concept without the need for accurate actuation for configuration-switching. Note that although the topic is very interesting, it is technically too specific for a general journal and most importantly, the adopted design method is well developed, and the quality of the paper does not meet the high standard of Nature Communications. The paper is thus not recommended for publication in Nature Communications.

The following comments can be considered by the authors when revising their paper:

(1) The authors presented a study of easy-to-actuate multi-compatible truss structures with prescribed reconfiguration. They adopted a two-step method, i.e., structure optimization followed by optimizing actuation. A better strategy would be formulating a single optimization problem with both prescribed reconfigurations and actuation as constraints.

(2) The authors presented an inverse design method for multi-compatible truss structures. Two examples are given. The first is a two-plate-four-bar structure (2P4B), and the second is a three-plate-six-bar (3P6B) structure. Although these two examples are already relatively simple, self-contact between components emerges. The component interference is expected to increase dramatically with the complexity of the structure. However, it seems that the authors could only cope with the interference in an ad hoc manner. In fact, they were only able to show the experimental results for the simpler example (i.e., 2P4B). Without taking the interference into account in the design stage, the obtained results by the method may not be as useful as one expects if much more complex structures are preferred.

(3) The authors adopted the nudged elastic band method (NEB) to obtain the minimum energy reconfiguration path for the multi-compatible truss structures. Note that this method is well-developed in computational chemistry. Details of the method, e.g., Eqs. (3–6) and the associated texts, can be moved from the main text to Supplementary Information.

Response to Reviewers

The reviewers' comments are highly insightful and enabled us to greatly improve our manuscript. A major improvement is the inclusion of a quadra-stable gripper with variable stiffness (in Fig. 7 and Supplementary Video 5) to demonstrate the practical usefulness of this design method. Our point-by-point responses to each comment are shown below. Text revisions in the manuscript are shown in red font, and modified figures are boxed by red frames.

Reviewer #1

General Comment: The submitted manuscript titled "Easy-to-actuate multi-compatible truss structures with prescribed reconfiguration" presents a novel method to synthesize multistable mechanisms that eliminate the redundant stable configurations to a large extent. Moreover, the manuscript develops a method to design simple approximate actuation by determining the minimum energy path using the nudged elastic band method. To my knowledge, generating precise, stable configurations using multi-compatibility constraints is novel and provides a new way of developing lightweight multistable mechanisms. The manuscript further explains how MEPs can be used to design efficient inaccurate actuators. The manuscripts say, "NEB widely used in computational chemistry to explore MEP." In the past five years, some manuscripts have introduced MEP to solve MEP problems in nonlinear structural mechanics. The authors should clearly state that. However, using potential landscape exploration algorithms to obtain an approximate but effective actuation is of great value for designers seeking to develop actuation systems for complex nonlinear mechanisms. The manuscript is well written, and ideas have been communicated well. Further, the manuscript provides sufficient evidence supporting the claims made in the conclusion.

Response: Thank you for the positive comments about the two parts of our work ("multi-compatible truss design" and "actuation design based on the MEP found by NEB"), and the authors feel encouraged and deeply grateful. Thank you for pointing out the value of designing an actuation, we added the following sentences to the corresponding position in Page 4:

Page 4 Line 139: and introduced to the nonlinear structural mechanics to calculate MEP in the recent years²⁷⁻³⁰

Comment 1: Page 1 and 2: The authors have made certain general remarks that most multistable literature structures are assembled bistable unit cells. This remark is not valid, as there are several examples of multistable structures that do not constitute bistable repeating cells:

a. Risso G, Ermanni P. Multi-stability of fiber reinforced polymer frames with different geometries. *Composite Structures*. 2023 Jun 1;313:116958.

b. Pirrera A, Lachenal X, Daynes S, Weaver PM, Chenchiah IV. Multistable cylindrical lattices. *Journal of the Mechanics and Physics of Solids*. 2013 Nov 1;61(11):2087-107.

Response: Thank you for supplementing the literature review. Except for multistable structures assembled by bistable unit cells, a separate point is included for the other design methods with their advantages and limitations in the introduction part in Page 2. Accordingly, we added the following sentences in Page 1 to make it more rigorous and supplemented the literature that the reviewer mentioned here in the introduction in Page 2:

Page 1 Line 19: Another type is by introducing prestress together with kinematic symmetry or nonlinearity to

achieve multi-stability, but the accurate implementation of prestress is difficult, and the resultant structure often suffers the lack of stiffness.

Page 2 Line 55: *Bi/multi-stability can also be emerged from prestressed shells, such as bistable tape springs¹², helical connection of pre-curved shell strips¹³, and pre-tensioning shell frames for multiple out-of-plane buckling modes¹⁴. Such multi-stable compliant shell structures also have many distracting stable states and need many actuators to control¹⁵.*

12. Wang, B., Seffen, K. A., & Guest, S. D. *Folded strains of a bistable composite tape-spring. International Journal of Solids and Structures* 233, 111221 (2021).

13. Pirrera, A., Lachenal, X., Daynes, S., Weaver, P. M., & Chenchiah, I. V. *Multi-stable cylindrical lattices. Journal of the Mechanics and Physics of Solids* 61, 2087-2107 (2013).

14. Risso, G., & Ermanni, P. *Multi-stability of fiber reinforced polymer frames with different geometries. Composite Structures* 313, 116958 (2023).

15. Risso, G., Sakovsky, M., & Ermanni, P. *A Highly Multi-Stable Meta-Structure via Anisotropy for Large and Reversible Shape Transformation. Advanced Science* 9, 2202740 (2022).

Comment 2: As pointed out earlier, the literature on structural mechanics has shown MEP using methods like nudged elastic band, dimer, and conjugate peak refinement. Further, it is possible to tailor the energy barriers with alternate methods of producing multistable structures.

Response: Thank you for the insightful comment. This has greatly helped us to improve the quality of our manuscript. As shown in Fig. 3, we attempted to use local stiffness to influence the energy barrier. We can observe that it is effective for this 2P4B structure, but it may not ensure effective control of the energy barrier. Therefore, based on the prediction of the energy barriers by the NEB method, we added an algorithm that incorporates the iterative process of NEB into the step of the multi-compatibility design so that to realize tailoring the energy barriers in the meantime produce multi-stable structures:

Page 6 Line 159: *.....A higher-level loop has been applied on top of the three-step optimization for a more compact actuation design and **energy barrier control**. The detailed algorithms are provided in Supplementary Information, Section 5.*

SI Section 5: *The multi-compatible structure can be designed with many other constraints or targets, such as tailoring the energy barrier or together with the actuation design for a more suitable actuator. However, achieving these further targets requires information on the MEP calculated by the NEB method, an iteration procedure. Therefore, we constructed a loop algorithm with the warm-start optimization, where the results obtained from a previous optimization are used as the initial guess or starting point for a subsequent optimization process. Generate a preliminary multi-stable structure by the first optimization Opt.1 and employ the NEB method to obtain the MEP. Then input this preliminary designed structure as the initial guess to the second optimization Opt.2, whose constraints based on the calculated MEP are added to the multi-compatibility constraints. The loop algorithms for the entire easy-to-actuate multi-stable structure design and energy barrier tailoring are shown in Supplementary Fig. 7. [6]*

[6] Jingyi Yang et al. "Folding and deploying identical thick panels with spring-loaded hinges". In: *Extreme Mechanics Letters* 52 (2022), p. 101637.

SI Section 5.2 Algorithm for tailoring the energy barrier

In Supplementary Fig. 7b, generate a multi-stable structure G1 that can achieve prescribed configurations and

include the local stiffness in the constraints to guide the energy barriers to satisfy the energy tailoring target in Opt.1 (see Supplementary Information, Section 3 for the discussion about the relationship between the local stiffness and the energy barrier in detail). Calculate its reconfiguration path MEP 1 through the NEB method. Check whether the results of Opt.1 with local stiffness design satisfy the energy tailoring condition. If it fails, input the structure and stiffness design G1, K1 as the initial guess of another optimization Opt.2, which satisfies the multi-compatibility constraints and the energy tailoring constraints based on MEP 1. Then check whether the designs G2, K2, MEP2 from Opt.2 meet the energy condition. If it works, a multi-stable structure with prescribed energy barriers is obtained; if it does not work, take the MEP of the Opt. 2 results as the input parameters to carry out the second optimization again, until the output MEP satisfies the energy target.

Supplementary Fig. 9 demonstrates the results of Opt.1 and Opt.2 respectively. The target of energy tailoring is that the energy barrier of the reconfiguration from Config. 2 to 3 E23 is lower than that from Config. 1 to 2 E12. Supplementary Fig. 9a presents that the results of the first optimization Opt. 1 did not satisfy the energy barrier tailoring target. Supplementary Fig. 9b demonstrates the final design results of the Opt. 2 satisfied the target. Furthermore, the same optimization Opt.3 as Opt.2 can be performed more times on the successful results G2, K2, MEP2 until a more high-standard condition for the energy barriers converges, such as the difference between E23 and E12 no longer decreases.

Supplementary Fig. 7. **The flow chart of the warm-start optimization, where the results obtained from a previous optimization are used as the initial guess or starting point for a subsequent optimization process. a** The warm-start optimization for the actuation design. **b** The warm-start optimization for tailoring energy barriers.

Supplementary Fig. 9. **The result of the double-optimization for tailoring the energy barrier.** The target of energy tailoring is that the energy barrier of the reconfiguration from Config. 2 to 3 is lower than that from Config. 1 to 2. **a** The result from Opt. 1. Although the local stiffness is considered in the constrained equations, the result did not satisfy the prescribed target of the energy barrier. **b** According to the MEP obtained from the design from Opt. 1, both the geometry of the structure and the stiffness of components are optimized to satisfy the target of energy tailoring.

Comment 3: The work in the manuscript is at the intersection of robotics (rigid-body mechanics) and continuum mechanics. Hence, it is essential to provide references to seemingly prominent concepts like the Chebychev-Grubler-Kutzback (CGK) criterion for better readability.

Response: Thank you for the insightful suggestion. A reference is added with the text mentioning the Chebychev-Grubler-Kutzback (C-G-K) criterion in Page 3:

Page 3 Line 102: *thus immovable according to the Chebychev–Grübler–Kutzback (C-G-K) criterion²⁴.*

24.Hunt, K. H. Kinematic Geometry of Mechanisms (Oxford Univ. Press, London, 1978).

Comment 4: Fig. 1 explains how to build multi-compatible truss structures. In Fig. 1B, please explicitly show that the incorporated structure has a common proximal link. Moreover, the manuscript identifies two additional intersection points, U and N (which point to additional stable states apart from the three target equilibrium configurations). What makes the intersection point U the only compatible point needs to be clarified in the text.

Response: Thank you for the suggestion. We refined the common proximal link of the incorporated structure in Fig. 1b and added a subfigure to illustrate the intersection points U and N. A more detailed explanation both in the figure legend and main text as follows:

Figure. 1b(III): *.....There are two more path intersections in the 2D plot, U and N. When considering the rotation of the coupler link to plot trajectories in 3D space in the blue dashed box, only the point U is an intersection, i.e. a compatible state.*

Page 3 Line 109: When considering the rotation of the coupler link, only the point U is an intersection, i.e. a compatible state, in the 3D plot as shown in the blue dashed box in Fig. 1b(III).

Comment 5: Check Eq. 5 and 6. There could be an issue with the Microsoft Word version. Eq. 6 needs to be printed correctly.

Response: Thank you for the feedback. We submitted the PDF version this time to avoid the software version problem, and we will proof the printing problem with the editor finally.

Comment 6: From the initial description of Fig. 1C, it needs to be clarified what Config Space: PC1 and Config Space PC2 mean. Please clarify the axes in the text.

Response: Thank you for this suggestion. We added the clarification of the axes in Page 4 Line 141.

Page 4 Line 141: Fig. 1c shows the principle of the NEB method in the configuration space after the dimensionality reduction by the Principal Component Analysis (PCA) method, where C_1 and C_N represent the configuration points of the two respective stable states that are designed.

Comment 7: Page 10 states that the MEP is plotted under Principal Component Analysis (PCA) projection. This line explains Point 6 to an extent. However, the details of the transformation between the relevant degrees of freedom should be included in the Supplementary information document.

Response: Thank you for this suggestion. The details of the dimensionality reduction by applying PCA are added in the Supplementary Information (SI) Section 11.2.

Page 10 Line 241: Fig. 4a demonstrates the MEP of the 2P4B structure under PCA projection (see Supplementary Information, Section 11.2 for discussions on the transformation between the relevant degrees of freedom),.....

SI Section 11.2: To plot the high dimensional minimum energy path (MEP) in three dimensions, the dimensionality reduction of the data is conducted by the Principal Component Analysis (PCA) method, and MATLAB's built-in function "pca" is used. This section takes the 2P4B structure as an example to illustrate the transformation between the relevant degrees of freedom. The original data of the MEP from Config. 1 to Config. 2 of the 2P4B structure as shown in the black curve in Fig. 4a is written as Eq. (58).

$$CFG = \begin{bmatrix} 2.25 & -0.02 & 2.74 & 0.08 & 3.00 & 0.51 & 2.86 & 0.99 & 0.39 & -0.92 & 0.81 & -0.58 & 1.00 & 0.06 & 0.84 & 0.55 & 0 & 0 \\ 2.25 & -0.02 & 2.74 & 0.08 & 3.00 & 0.51 & 2.86 & 0.99 & 0.36 & -0.84 & 0.79 & -0.51 & 0.98 & 0.13 & 0.82 & 0.62 & -0.02 & 0.08 \\ \vdots & \vdots & \vdots & \vdots & \vdots & \vdots & \vdots & \vdots & \vdots & \vdots & \vdots & \vdots & \vdots & \vdots & \vdots & \vdots & \vdots & \vdots \\ 2.25 & -0.02 & 2.74 & 0.08 & 3.00 & 0.51 & 2.86 & 0.99 & 1.88 & 2.01 & 2.42 & 2.09 & 2.90 & 2.56 & 3.00 & 3.06 & 2 & 3 \end{bmatrix} \quad (58)$$

where CFG is a 52×18 matrix, in which one row represents one configuration on the MEP and the number of rows denotes how many discrete configurations form the continuous MEP. The number of elements in one row illustrates the DOF of the structure.

Then input the matrix CFG as the feature matrix to the MATLAB's built-in function "pca", a coefficient matrix ce that records the coefficients for each principal component (PC) and a matrix sc that records the coordinates of the feature matrix in the PC space are obtained.

$$ce = \begin{bmatrix} 0.00 & 0.00 & 0.00 & 0.00 & 0.00 & 0.00 & 0.00 & 0.00 & 0.00 & 0.00 & 0.75 & -0.25 & 0.58 & -0.18 & -0.01 & 0.00 & 0.00 & 0.00 \\ 0.00 & 0.00 & 0.00 & 0.00 & 0.00 & 0.00 & 0.00 & 0.00 & 0.00 & 0.00 & 0.00 & 0.91 & 0.30 & -0.27 & -0.03 & 0.00 & 0.00 & 0.00 \\ 0.00 & 0.00 & 0.00 & 0.00 & 0.00 & 0.00 & 0.00 & 0.00 & 0.00 & 0.00 & 0.38 & 0.30 & -0.10 & 0.86 & 0.12 & 0.00 & 0.00 & 0.00 \\ 0.00 & 0.00 & 0.00 & 0.00 & 0.00 & 0.00 & 0.00 & 0.00 & 0.00 & 0.00 & 0.00 & 0.00 & -0.03 & -0.14 & 0.99 & 0.01 & 0.00 & 0.00 \\ 0.00 & 0.00 & 0.00 & 0.00 & 0.00 & 0.00 & 0.00 & 0.00 & 0.00 & 0.00 & -0.38 & -0.10 & 0.53 & 0.26 & 0.04 & 0.71 & -0.04 & 0.00 \\ 0.00 & 0.00 & 0.00 & 0.00 & 0.00 & 0.00 & 0.00 & 0.00 & 0.00 & 0.00 & 0.00 & 0.00 & 0.00 & 0.00 & 0.00 & 0.05 & 0.95 & 0.32 \\ 0.00 & 0.00 & 0.00 & 0.00 & 0.00 & 0.00 & 0.00 & 0.00 & 0.00 & 0.00 & 0.38 & 0.10 & -0.53 & -0.25 & -0.06 & 0.71 & -0.04 & 0.00 \\ 0.00 & 0.00 & 0.00 & 0.00 & 0.00 & 0.00 & 0.00 & 0.00 & 0.00 & 0.00 & 0.00 & 0.00 & 0.00 & 0.00 & 0.00 & -0.01 & -0.32 & 0.95 \\ 0.21 & 0.38 & 0.50 & 0.25 & -0.11 & -0.20 & 0.65 & -0.01 & -0.15 & -0.03 & 0.00 & 0.00 & 0.00 & 0.00 & 0.00 & 0.00 & 0.00 & 0.00 \\ 0.40 & -0.24 & -0.25 & 0.50 & -0.18 & -0.03 & 0.03 & 0.54 & 0.31 & -0.23 & 0.00 & 0.00 & 0.00 & 0.00 & 0.00 & 0.00 & 0.00 & 0.00 \\ 0.22 & 0.37 & 0.33 & -0.03 & 0.00 & -0.04 & -0.37 & -0.23 & 0.71 & -0.08 & 0.00 & 0.00 & 0.00 & 0.00 & 0.00 & 0.00 & 0.00 & 0.00 \\ 0.36 & -0.24 & 0.08 & 0.30 & 0.57 & -0.24 & -0.13 & -0.16 & -0.09 & 0.53 & 0.00 & 0.00 & 0.00 & 0.00 & 0.00 & 0.00 & 0.00 & 0.00 \\ 0.26 & 0.36 & -0.08 & -0.29 & -0.35 & -0.01 & -0.19 & 0.42 & -0.14 & 0.60 & 0.00 & 0.00 & 0.00 & 0.00 & 0.00 & 0.00 & 0.00 & 0.00 \\ 0.34 & -0.25 & 0.33 & -0.11 & -0.29 & -0.29 & -0.44 & -0.08 & -0.45 & -0.36 & 0.00 & 0.00 & 0.00 & 0.00 & 0.00 & 0.00 & 0.00 & 0.00 \\ 0.29 & 0.37 & -0.42 & -0.35 & 0.45 & -0.34 & 0.08 & 0.10 & -0.09 & -0.37 & 0.00 & 0.00 & 0.00 & 0.00 & 0.00 & 0.00 & 0.00 & 0.00 \\ 0.34 & -0.26 & 0.34 & -0.43 & 0.28 & 0.58 & 0.19 & 0.24 & 0.05 & -0.08 & 0.00 & 0.00 & 0.00 & 0.00 & 0.00 & 0.00 & 0.00 & 0.00 \\ 0.27 & 0.38 & -0.21 & 0.37 & 0.01 & 0.60 & -0.18 & -0.29 & -0.33 & -0.11 & 0.00 & 0.00 & 0.00 & 0.00 & 0.00 & 0.00 & 0.00 & 0.00 \\ 0.40 & -0.25 & -0.35 & -0.23 & -0.38 & -0.03 & 0.34 & -0.54 & 0.18 & 0.13 & 0.00 & 0.00 & 0.00 & 0.00 & 0.00 & 0.00 & 0.00 & 0.00 \end{bmatrix} \quad (59)$$

where ce is an 18×18 matrix, in which each column contains coefficients for one principal component and the columns are in descending order of component variance. The matrix sc is also a 52×18 matrix, where each row corresponds to a row in the original data matrix CFG and each column corresponds to a principal component. Therefore, to plot the Path 1-2 (i.e. the MEP from Config. 1 to Config. 2) in three dimensions as shown in Fig. 4a, the first two columns are extracted from the matrix sc as the x-axis and y-axis of the principal component space as Eq. (60), the z-axis is filled with the values of the physical energy corresponding to 52 configurations on the MEP.

$$sc(:, 1 : 2)' = \begin{bmatrix} -3.67 & 1.12 \\ -3.56 & 1.00 \\ -3.45 & 0.87 \\ -3.33 & 0.75 \\ -3.21 & 0.64 \\ -3.09 & 0.53 \\ -2.96 & 0.42 \\ -2.83 & 0.32 \\ -2.69 & 0.23 \\ -2.56 & 0.14 \\ -2.42 & 0.05 \\ -2.27 & -0.03 \\ -2.13 & -0.11 \\ -1.98 & -0.18 \\ -1.83 & -0.24 \\ -1.68 & -0.30 \\ -1.53 & -0.36 \\ -1.37 & -0.41 \\ -1.21 & -0.45 \\ -1.05 & -0.49 \\ -0.90 & -0.53 \\ -0.73 & -0.55 \\ -0.57 & -0.58 \\ -0.41 & -0.59 \\ -0.25 & -0.61 \\ -0.09 & -0.61 \\ 0.08 & -0.61 \\ 0.24 & -0.61 \\ 0.40 & -0.60 \\ 0.56 & -0.58 \\ 0.72 & -0.56 \\ 0.88 & -0.53 \\ 1.04 & -0.50 \\ 1.20 & -0.46 \\ 1.36 & -0.42 \\ 1.52 & -0.37 \\ 1.67 & -0.31 \\ 1.83 & -0.25 \\ 1.98 & -0.18 \\ 2.13 & -0.11 \\ 2.27 & -0.04 \\ 2.42 & 0.04 \\ 2.56 & 0.13 \\ 2.70 & 0.22 \\ 2.84 & 0.32 \\ 2.97 & 0.42 \\ 3.10 & 0.53 \\ 3.23 & 0.64 \\ 3.35 & 0.76 \\ 3.47 & 0.88 \\ 3.58 & 1.00 \\ 3.69 & 1.13 \end{bmatrix}, \quad E_{pny} = \begin{bmatrix} 0.00E+00 \\ 2.13E-05 \\ 8.11E-05 \\ 1.74E-04 \\ 2.95E-04 \\ 4.40E-04 \\ 6.05E-04 \\ 7.84E-04 \\ 9.78E-04 \\ 1.17E-03 \\ 1.37E-03 \\ 1.57E-03 \\ 1.77E-03 \\ 1.97E-03 \\ 2.16E-03 \\ 2.34E-03 \\ 2.52E-03 \\ 2.68E-03 \\ 2.83E-03 \\ 2.96E-03 \\ 3.08E-03 \\ 3.18E-03 \\ 3.26E-03 \\ 3.33E-03 \\ 3.38E-03 \\ 3.40E-03 \\ 3.41E-03 \\ 3.40E-03 \\ 3.37E-03 \\ 3.31E-03 \\ 3.24E-03 \\ 3.16E-03 \\ 3.05E-03 \\ 2.92E-03 \\ 2.78E-03 \\ 2.63E-03 \\ 2.46E-03 \\ 2.28E-03 \\ 2.09E-03 \\ 1.89E-03 \\ 1.69E-03 \\ 1.48E-03 \\ 1.27E-03 \\ 1.06E-03 \\ 8.58E-04 \\ 6.68E-04 \\ 4.92E-04 \\ 3.34E-04 \\ 2.01E-04 \\ 9.82E-05 \\ 3.06E-05 \\ 0.00E+00 \end{bmatrix} \quad (60)$$

Comment 8 (regarding the supplementary information document): Overall, the supplementary information document is also well compiled. Considering the manuscript's multi-disciplinary nature, the authors should include a step-by-step detailed construction of at least the 2P4B with three prescribed conditions. It is strongly recommended that the authors include the working code of one example as a part of the supplementary material.

It is strongly recommended that the PCA be applied to one of the examples to construct the MEP plots, as shown.

Response: Thank you for this suggestion to help us improve the quality of the paper. A step-by-step detailed construction of the 2P4B structure is added in Supplementary Information (SI) Section 11.4. The working code of the 2P4B structure is attached to Supplementary Code 1. The design toolbox for customizing the multi-stable plate-and-bar structure is attached to Supplementary Code 2.

Page 6 Line 170: (See Supplementary Information, Section 11.4 for the step-by-step design procedure of the 2P4B structure.)

We solve the optimization problem numerically using MATLAB's built-in optimization routine "fmincon" (see Supplementary Code 1 for the 2P4B structure design, Supplementary Code 2 for the design toolbox for customizing the multi-stable plate-and-bar structure.)

SI Section 11.4: The design procedure of a simple multi-stable 2P4B structure is demonstrated step-by-step. In two dimensions, bodies are constrained in a plane to be plates. So the structure consists of six plates, $nb=6$, as shown in Supplementary Fig. 22a. This structure was designed with three stable configurations and a given topology. The only design variables are the coordinates of all joints A . All computations were carried out with MATLAB(2020).

Supplementary Fig. 22 **The design procedure of the multi-stable 2P4B structure.** **a** Notations of the variables on the structure. **b** Plates 2-5 can be simplified to bars when only prescribed the target configurations of plate 1 and plate 6 and that of plates 2-5 are out of interest. **c** The result of the constrained optimization of the multi-compatibility structure design. **d** The corresponding physical model.

11.4.1 Step A: Prescribe information about each target configuration

Assume only plate 1 and plate 6 are of interest, which means only reference points and rotation angles of these two plates are prescribed and others are set as variables. So neglecting reference points of plate 2 to plate 5, these plates are simplified as bars, as shown in Supplementary Fig. ??b. Then that two plates are connected by four bars is the topology in this example. Only the poses (reference points and rotation angles) of these plates are of interest, one plate was fixed at ground and its reference points at three target configurations were set as $[0,0]$, its rotation angles were set as 0. The reference points of the other plate were prescribed at three target configurations, $nc=3$, shown in Eq. (61),(62).

$$P = \begin{bmatrix} P_1^1 \\ P_2^1 \\ P_3^1 \\ P_1^6 \\ P_2^6 \\ P_3^6 \end{bmatrix} = \begin{bmatrix} 0, 0 \\ 2, 3 \\ 4, 1 \\ 0, 0 \\ 0, 0 \\ 0, 0 \end{bmatrix} \quad (61)$$

$$\theta = \begin{bmatrix} \theta_1^1 \\ \theta_2^1 \\ \theta_3^1 \\ \theta_1^6 \\ \theta_2^6 \\ \theta_3^6 \end{bmatrix} = \begin{bmatrix} 0^\circ \\ 30^\circ \\ 5^\circ \\ 0^\circ \\ 0^\circ \\ 0^\circ \end{bmatrix} \quad (62)$$

11.4.2 Step B: Conduct constrained optimization on the geometric configuration

For two dimension, the rotation matrix should be:

$$Rot(\bullet) = \begin{bmatrix} \cos \theta & -\sin \theta \\ \sin \theta & \cos \theta \end{bmatrix} \quad (63)$$

The rotation equations of these six components from configuration 1 to configuration 2 and 3 could be written. The target configurations of body 1 are prescribed as Eq. (61)-(62), the rotation equations of it are:

$$\begin{aligned} \mathbf{A}_2^{1,p} - \begin{bmatrix} 2 \\ 3 \end{bmatrix} &= \begin{bmatrix} 0.8660 & -0.5000 \\ 0.5000 & 0.8660 \end{bmatrix} \cdot (\mathbf{A}_1^{1,p} - \begin{bmatrix} 0 \\ 0 \end{bmatrix}) \\ \mathbf{A}_3^{1,p} - \begin{bmatrix} 4 \\ 1 \end{bmatrix} &= \begin{bmatrix} 0.9962 & -0.0872 \\ 0.0872 & 0.9962 \end{bmatrix} \cdot (\mathbf{A}_1^{1,p} - \begin{bmatrix} 0 \\ 0 \end{bmatrix}) \end{aligned} \quad (64)$$

where p equals to 2-5 because body 1 is connected with body 2, 3, 4, 5. Then all A points at other configurations could be expressed by all A points at configuration 1 and only A points at configuration 1 are regarded as variables. Above equations could be transformed into following forms:

$$\begin{aligned} \mathbf{A}_2^{1,p} &= \begin{bmatrix} 2 \\ 3 \end{bmatrix} + \begin{bmatrix} 0.8660 & -0.5000 \\ 0.5000 & 0.8660 \end{bmatrix} \cdot (\mathbf{A}_1^{1,p} - \begin{bmatrix} 0 \\ 0 \end{bmatrix}) \\ \mathbf{A}_3^{1,p} &= \begin{bmatrix} 4 \\ 1 \end{bmatrix} + \begin{bmatrix} 0.9962 & -0.0872 \\ 0.0872 & 0.9962 \end{bmatrix} \cdot (\mathbf{A}_1^{1,p} - \begin{bmatrix} 0 \\ 0 \end{bmatrix}) \end{aligned} \quad (65)$$

Considering the poses of plate 2 to plate 5 are of no interest, the reference points and rotation angles of these plates are regarded as variables, their rotation equations could be written as follows:

$$\mathbf{A}_i^{p,q} - \mathbf{P}_i^p = \begin{bmatrix} \cos \theta_i^p & -\sin \theta_i^p \\ \sin \theta_i^p & \cos \theta_i^p \end{bmatrix} \cdot (\mathbf{A}_1^{p,q} - \mathbf{P}_1^p) \quad (66)$$

where $p=2,\dots,5$, $i=2, 3$, because components 2-5 are all connected between plate 1 and plate 6, so q equals to 1 and 6. P_i^p , θ_i^p and $A_1^{p,q}$ are variables and $A_i^{p,q}$ could be expressed by above variables as follows:

$$\mathbf{A}_i^{p,q} = \mathbf{P}_i^p + \begin{bmatrix} \cos\theta_i^p & -\sin\theta_i^p \\ \sin\theta_i^p & \cos\theta_i^p \end{bmatrix} \cdot (\mathbf{A}_1^{p,q} - \mathbf{P}_1^p) \quad (67)$$

Because the plate 6 was fixed at the ground, there is no need to write its rotation equations.

Above rotation equations ensure the size of all components unchanged at three target configurations. The condition about all joints as Eq. (11) were written as equality constraints:

$$C_{eq_{cpt}}(\mathbf{A}) = \left\{ \begin{array}{l} |\mathbf{A}_i^{1,q1} - \mathbf{A}_i^{q1,1}| = 0, \quad q1 = 2, 3, 4, 5 \\ |\mathbf{A}_i^{6,q6} - \mathbf{A}_i^{q6,6}| = 0, \quad q6 = 2, 3, 4, 5 \end{array}, \quad i = 1, 2, 3 \right\} \quad (68)$$

Except for fundamental two conditions, two categories of inequality constraints were added to limit the size of these components, one category of inequality constraints describes the distance between the reference point and the joint points, presented as follows:

$$C_{size}(\mathbf{A}) = \left\{ \begin{array}{l} 1 \leq |\mathbf{P}_1^1 - \mathbf{A}_1^{1,q1}| \leq 10, \quad q1 = 2, 3, 4, 5 \\ 1 \leq |\mathbf{P}_1^6 - \mathbf{A}_1^{6,q6}| \leq 10, \quad q6 = 2, 3, 4, 5 \end{array} \right\} \quad (69)$$

Another category of inequality constraints limits the distance among joints, the lower bound setting prevents these joints overlapping, the upper bound setting limits the size of bodies, shown as follows:

$$C_{size}(\mathbf{A}) = \left\{ \begin{array}{l} 1 \leq |\mathbf{A}_1^{p,2} - \mathbf{A}_1^{p,3}| \leq 10 \\ 1 \leq |\mathbf{A}_1^{p,2} - \mathbf{A}_1^{p,4}| \leq 10 \\ 1 \leq |\mathbf{A}_1^{p,2} - \mathbf{A}_1^{p,5}| \leq 10 \\ 1 \leq |\mathbf{A}_1^{p,3} - \mathbf{A}_1^{p,4}| \leq 10 \\ 1 \leq |\mathbf{A}_1^{p,3} - \mathbf{A}_1^{p,5}| \leq 10 \\ 1 \leq |\mathbf{A}_1^{p,4} - \mathbf{A}_1^{p,5}| \leq 10 \end{array}, p = 1, 6 \right\} \quad (70)$$

Plate 2 to 5 were simplified as bars without reference points, so only the second term in Eq. (15) is considered for plate 2 to 5:

$$\begin{aligned} E(\mathbf{A}) = & \sum_{p=2,3,4,5} |\mathbf{A}_1^{p,1} - \mathbf{A}_1^{p,6}| + \sum_{q1=2,3,4,5} |\mathbf{P}_1^1 - \mathbf{A}_1^{1,q1}| + \sum_{q6=2,3,4,5} |\mathbf{P}_1^6 - \mathbf{A}_1^{6,q6}| \\ & + \sum_{p=1,6} (|\mathbf{A}_1^{p,2} - \mathbf{A}_1^{p,3}| + |\mathbf{A}_1^{p,2} - \mathbf{A}_1^{p,4}| + |\mathbf{A}_1^{p,2} - \mathbf{A}_1^{p,5}| + |\mathbf{A}_1^{p,3} - \mathbf{A}_1^{p,4}| \\ & + |\mathbf{A}_1^{p,3} - \mathbf{A}_1^{p,5}| + |\mathbf{A}_1^{p,4} - \mathbf{A}_1^{p,5}|) \end{aligned} \quad (71)$$

11.4.3 Solutions

The options of the nonlinear optimization solver "fmincon" were set as default. The results of the optimization are shown in Tab. 1 and Fig. 22c. Because plate 6 was fixed at the ground, the coordinates of joints were the same at different target configurations. In Fig. 22c, the grey part means component 6 fixed at the ground while the other three plates with colors denote component 1 at three target configurations, and bars between two plates represent components 2 to 5.

Config.(i)	$\mathbf{A}_i^{1,2}$	$\mathbf{A}_i^{1,3}$	$\mathbf{A}_i^{1,4}$	$\mathbf{A}_i^{1,5}$
1	[0.3949,-0.9187]	[0.8144,-0.5803]	[0.9980,0.0635]	[0.8359,0.5488]
2	[1.8826,2.0069]	[2.4152,2.0903]	[2.8960,2.5526]	[2.9984,3.0573]
3	[4.3133,0.0504]	[4.7607,0.3509]	[4.9997,0.9763]	[4.8806,1.4739]
Config.(i)	$\mathbf{A}_i^{6,2}$	$\mathbf{A}_i^{6,3}$	$\mathbf{A}_i^{6,4}$	$\mathbf{A}_i^{6,5}$
1	[2.2522,-0.0221]	[2.7418,0.0795]	[3.0019,0.5065]	[2.8638,0.9870]
2	[2.2522,-0.0221]	[2.7418,0.0795]	[3.0019,0.5065]	[2.8638,0.9870]
3	[2.2522,-0.0221]	[2.7418,0.0795]	[3.0019,0.5065]	[2.8638,0.9870]

Table 1: 2D example solution from MATLAB

Reviewer #2

General Comment: In recent years, multi-stable structures have attracted the interest of scientists and engineers due to their special variable configurations and tunable performance. In their work, the authors proposed a method by a series of reference points to design multi-stable structures with numerous configurations. More importantly, the nudged elastic band method has been employed to understand the fundamental principle of the minimum energy path connecting the specified stable states. After that, some multi-stable structures were designed and tested to validate the effectiveness of the proposed method.

Response: We would like to thank this reviewer for the valuable comments and for the time and dedication in reviewing our manuscript.

Comment 1: The authors declared in the abstract that bi-stable unit cells contain multitudes of additional and distractive stable states (lines 16-18). I agree with this statement. However, the examples of multi-stable structures given in this work also include some units that will have unstable behaviors during the process of configuration change. Is the direction of instability of these units controllable, or is it random?

Response: Thank you for the sharp comments. There are two categories of examples, one is the overall multi-stable structures as shown in Fig. 2a and 2c (i.e. 2P4B and 3P6B), which are overall stable and can be easily actuated as shown in Supplementary Video 1 and 2. With the complexity of the structure increasing, the 3P6B structure has two additional configurations, which are out of our expected stable states but can be predicted by the reconfiguration path finding in advance as shown in Fig. 4c. And the final physical model worked as the prediction in Fig. 4c as shown in Supplementary Video 2.

The other examples are the four-unit assembly in Fig. 2g and the three-unit assembly in Fig. 3c, aiming to demonstrate that the multi-stable structures in this paper have the potential to be unit cells for more complex applications, e.g. metamaterials. If one unit cell has n_s stable states and the assembly consists of n_u unit cells, the assembly will have $n_s * n_u$ stable states in total. Therefore, when the assembly is expected to reconfigure between two of these $n_s * n_u$ stable states, it is possible that the assembly would be trapped in the other stable states. However, similar to the 3P6B example in Fig. 4c, the reconfiguration path can be calculated by the NEB method and the actuator can be designed to drive the structure to reconfigure along the calculated path. The following paragraph is added to SI Section 10.3. In the future, the stability of each unit cell can be designed to avoid additional stable states on the reconfiguration path.

Page 10 Line 266: *As a result, we find that computing MEP and correspondingly designing the actuation are crucial in the successful navigation of the desired reconfiguration when many diverging stable states are present (see Supplementary Information, Section 10.3 for applying the MEP finding and actuation design on a more complex structure assembled by two quadra-stable unit cells.)*

SI Section 10.3: *To explore the generality of the reconfiguration path finding and actuation design method, we studied whether a more complex structure assembled by quadra-stable unit cells is controllable during the reconfiguration. A structure assembled by two quadra-stable cells as shown in Supplementary Fig. 20a would have 16 stable states in total since each unit cell has 4 stable states and two of all stable states are prescribed as the target configurations for the assembly. The reconfiguration path between two target configurations is found by the NEB method as shown by the blue curve in Supplementary Fig. 20b, which predicted the structure preferred to experience 5 of all stable states to achieve the prescribed reconfiguration between two target configurations as shown in Supplementary Fig. 20c. Then the actuation design is conducted according to the found reconfiguration*

path. Supplementary Fig. 20d demonstrates the results of the simulation that actuates the assembled structure to reconfigure from Config. 1 to Config. 7 by the designed actuator denoted by the red dashed line, and the energy curve obtained by the simulation is presented by the purple curve in Supplementary Fig. 20b. We can observe that the reconfiguration process by the simulation (Supplementary Fig. 20)d matched well with the reconfiguration path calculated by the NEB method (Supplementary Fig. 20).

Supplementary Fig. 20 **The MEP of a two-unit-cell assembly, in which each unit cell has four stable states, is founded by the NEB method.** a Utilize two quadra-stable unit cells to assemble a multi-stable structure, the assembly would have 16 stable states in total for each unit cell has 4 stable states. Two of all stable states are prescribed as the target configurations. b The blue curve denotes the MEP founded by the NEB method, which predicted that the structure preferred to experience 5 of all stable states to achieve the prescribed reconfiguration from Config. 1 to 7. Then we conducted the actuation design, the actuator is shown as the red dashed line in c. The purple curve is the reconfiguration path through the simulation of the designed actuator. c, d The configurations on the reconfiguration path through the simulation matched well with those on the NEB path.

Comment 2: In line 99 shows that the pin-jointed bars in the model are deformable. Are their elastic deformations controllable or quantifiable?

Response: Thank you for the advice. In this paper, the MEP, as shown in Fig. 4a and c, calculated by the NEB method has provided information on the elastic deformations, because the MEP illustrates the variance of the strain energy of intermediate configurations between two prescribed stable states and the strain energy is calculated by the elastic deformations. It is noted that the physical strain energy calculated by the elastic deformations in the NEB method has different forms depending on the forms of the components, such as the truss model and the beam model. The introduction of the truss model and beam model is added to SI Section 4.1.

Page 9 Line 235: *We applied the truss model, i.e. each bar is a simple two-node linear member that only takes axial extension or compression, in the MEP finding for high efficiency (see Supplementary Information, Section 4.1 for the discussions on the truss model and beam model in detail.)*

SI Section 4.1: *The axial load versus axial displacement relationship is different for different elements. Here we discussed the truss element and the beam element. Truss elements are simple two-node linear members that only take axial extension or compression. The stress of the elements can be defined as:*

$$t = k_t x \quad (32)$$

where k is the stiffness of the element, x is the axial displacement of the node. Then the nodal load can be obtained by projecting the stress by the equilibrium matrix in Supplementary Equation (29).

When the beam element is subjected to compression, it appears to bend. When the beam is subjected to tension, the beam cannot be stretched, which is approximated by a bar with high stiffness. The stress of beam elements described by the axial load versus axial displacement relationship can be formulated as:

$$t = \begin{cases} -P_{cr} \left(1 - \frac{x}{2l}\right), & x < 0 \\ k_b x, & x \geq 0 \end{cases} \quad (33)$$

where l is the original length of the element, k_b is set to 10^4 times the slope of the bending force formula, P_{cr} is the critical load, written in the form

$$P_{cr} = \frac{\pi^2}{l^2} EI \quad (34)$$

where E is Young's modulus, I is the moment of inertia. To avoid the sudden change of the stress in the beam when the axial displacement changes the sign, the piecewise function in Supplementary Equation (33) needs to be smoothly blended.

According to our experience, different models only influence the values of the energy rather than the trend of the MEP. Thus, the strain energy of the structure is calculated by simply regarding the components as the truss elements in the MEP finding procedure. To characterize and verify the difference, we added the beam element analytic model and the comparison of the calculation results between the truss model and the beam model in Supplementary Information (SI) Section 4.3 as follows.

SI Section 4.3: *For the 2P4B structure, we applied the beam model in the NEB method and compared the MEP obtained by the truss model and beam model as shown in Supplementary Fig. 5. Although the energy curves, i.e. energy barrier, are a little different, the deformed configurations between the stable states are identical. Besides, the convergence of the NEB by the beam model took longer than that by the truss model. For the actuation design, it is significant to obtain the middle configurations during the reconfiguration rather than calculate accurate energy values. Therefore, if the values of energy barriers are important, the beam model can be chosen, if only requires*

designing the actuation at a lower time cost, it would be better to apply the truss model in the MEP finding.

Supplementary Fig. 5. **The comparison of the MEP obtained by the truss model and beam model.** **a** The MEP obtained by the truss model. The strain of the Config. 4 is much larger, so the strain of the others is not obvious. **b** The MEP obtained by the beam model. **c** The energy curve obtained by integrating the experimental force curve in Fig. 5. The energy barrier in the physical experiment between the stable state 1 and 2 is higher, which matched better with the beam model since the elastic bars made of TPU of the prototype deform as beam elements.

Comment 3: The samples shown in Fig.2 F and G were fabricated by TPU and carbon fiber. Is the interface between TPU and carbon fiber well connected? The authors declare that the function of the carbon fiber is improving the stiffness. Will the stiffness of the bars affect the multi-stability of the structure?

Response: Thank you for bringing out these meaningful discussions. The connection of the interface between TPU and carbon fiber depends on several conditions, such as the dryness of the material, the environment temperature, etc. To avoid the split between the two materials, we used TPU to wrap the rigid body made of carbon fiber. The description of the updated fabrication method is added to Supplementary Information, Section 8:

SI Section 8: In Supplementary Fig. 14d, the parts made of carbon fiber with high stiffness are fully wrapped by TPU to avoid the split between them, while the revolute joint is made of thin layers of TPU.

Supplementary Fig. 14. **Two methods of fabrication for the multi-stable structures.** One is 3D printing components and assembling them, as shown in a. The other is integrately 3D printing the whole physical model, as shown in c. **a** When the interference of components exists, layering is necessary. The components are 3D printed with PLA or TPU and assembled through the revolute joint fabricated, as shown in a. **b** The disassembly demonstration of a revolute joint. There is one bushing in each hole of the components. The shaft crosses two bushings forming a frictionless revolute joint, and two spacing rings are added to two endpoints of the shaft to prevent the shaft from slipping. **c** For the real-2D case, there is no interference of components. The physical model can be integrately printed. **d** The black layer is made of carbon fiber, and the white layer is made of TPU. The parts of carbon fiber are wrapper by TPU and the revolute joint is made of thin layers of TPU.

The stiffness of the carbon fiber is sufficiently large compared to TPU in Fig. 2f and 2g, so the carbon fiber bodies can be regarded as "stiff" components and the TPU hinges can be regarded as compliant components, which are similar to the relationship between PLA and TPU in Fig. 2b and 2d. Furthermore, the multi-stability is only related to the geometry designed by the multi-compatibility method. However, the stiffness of the bars would affect the local stiffness at stable states thus the energy barrier between stable states, which can be obtained in the local stiffness calculation in Supplementary Information Section 3. Thank the reviewer for raising this precious discussion, we added the discussion on the influence of the stiffness of the bars on the MEP, i.e. the energy variation curve during the reconfiguration.

SI Section 4.4: *To explore whether the stiffness of the bars affects the multi-stability of the structure, a parametric study about the influence of the stiffness of components on MEP is performed on the 2P4B structure. We set two values of stiffness, 1 and 10, representing elastic materials like TPU and stiff materials like PLA. By default, only replace part of four bars with TPU material to provide deformations. The result is presented in Supplementary Fig. 6, where 0 and 1 in the legend denote the bar is stiff (the stiffness is set to 10) or elastic (the stiffness is set to 1), and four numbers indicate the stiffness of the corresponding four bars. We can observe that the stiffness of the bars only affects the energy barrier rather than the multi-stability of the structure. The more elastic bars, the lower the energy barrier. When the number of elastic bars is identical, different positions of the elastic bars would also affect the energy barrier, as the comparison of the curves in warmer colors and colder colors.*

Supplementary Fig. 6. **A parametric study about the influence of the stiffness of components on the minimum energy path for the 2P4B structure.** In the legend, 0 and 1 denote the bar is stiff or elastic, and four numbers indicate the stiffness of the corresponding four bars.

Comment 4: The full name of the technical terms should be given before using the abbreviation (line 15).

Response: Thank you for pointing out this problem. Accordingly, the full name of the technical term is added and revisions in the manuscript are shown in red font.

Page 1 Line 16: *Consequently, multi-stable structures have the potential to achieve prescribed reconfiguration with only a few lightweight actuators (such as **shape-memory alloy** springs).*

Comment 5: “our-of-shelf” in section 7 in Supplementary material should be “out-of-shelf”.

Response: Thank you for pointing out this mistake. Accordingly, the word was corrected in Supplementary Information.

SI Section 8: *Because the bushing and the shaft are **out-of-shelf** parts, there is little friction in the rotation motion between them.*

Comment 6: References should be in a unified format. For example, the publication of reference 14 and the author information of reference 17 is missing.

Response: Thank you for pointing out the omission. We checked and revised the reference list thoroughly according to the formatting instructions from the editor.

Su, H. J., & McCarthy, J. M. Synthesis of bistable compliant four-bar mechanisms using polynomial homotopy. Journal of Mechanical Design 129, 1094-1098 (2007).

Dang, X., Feng, F., Duan, H., & Wang, J. Theorem on the compatibility of spherical kirigami tessellations. Physical Review Letters 128, 035501 (2022).

Comment 7: Is it possible to use temperature or magnetic field-sensitive raw materials to fabricate the proposed

structures so that they can automatically change configurations under different temperature or magnetic fields?

Response: Thank you for the insightful suggestion. We fully agreed the combination of the multi-stable structure and the external actuation field has a bright future. However, different from fabricating the components of the structure by the field-sensitive raw materials, we think it is better to add the actuator made of smart materials and use the external field to interact with the actuator to reconfigure the structure. This is because the multi-stable structure proposed in this paper satisfies multi-compatibility at the stable states, which means there is no deformation in the structure. Therefore, when the components fabricated by smart materials are actuated to deform from the original stable state, it is needed to remove the external field to recover the geometry of the components to enter the next stable state, but it is possible to return to the original state.

We intended to embed small magnets in the structure and apply magnetic field to reconfigure the structure, or use the SMA to fabricate the actuator in Fig. 4e and f in future work. The following sentence is added.

Page 12 Line 277: Furthermore, the actuation cable can be fabricated by the temperature or magnetic field-sensitive raw materials in the future so that the structure can automatically reconfigure under different external fields.

Reviewer #3

General Comment: The paper appears to be technically sound, proposing a problem, developing a solution, and demonstrating it.

The motivation of the work could be stronger. Background is given on lines 37-41 about the applications of other multi-stable structures, though it is unclear how this work fits within that context. Could it be applied generally to any of those applications, or is there a limited set to which the work applies? If this work does apply to any of those applications, what are the tradeoffs between designs? The paper would be strengthened by more there. Related, on lines 230-231, the authors mention that this could be used for a gripper with varying stiffness. On line 299, the authors mention it could be used for morphing wings or mechanically reconfigurable antennas. These are mentioned but it is not clear if or how it could be applied. What are the tradeoffs between this design and the work of others? The work would be significantly more compelling if there were a significant use case for the work.

Response: Thank you for the suggestions to improve and enrich our paper. This paper proposed a novel and promising design method for easy-to-actuate multi-stable structures based on the multi-compatibility constraints and MEP finding. We fully agree that applying this design method to at least one application will increase conviction of the practicality of the design method. Therefore, we added an application, a gripper with varying stiffness, by adopting a quadra-stable structure designed by the proposed method in this paper, and compared this design with the work of others, shown in Fig. 7 and Supplementary Video 5. The following paragraphs were added to Page 13.

Page 13 Line 288: *To explore the possible application of the proposed easy-to-actuate multi-stable structures, a variable stiffness gripper is designed employing a quadra-stable structure (Fig. 7a). It can reconfigure between 4 stable states with only 1 actuator. The reconfiguration between 2 (out of 4) stable states gives the stiff gripping mode, while the other 2 stable states provide the soft mode. One actuator is enough to operate the gripper in the two respective modes as well as switch between the two modes. Fig. 7b presents the specific force data of the gripper, where the gripping force in the stiff mode is around 30 times of the force in the soft mode. The measurement setup is presented in Supplementary Information, Section 9. To demonstrate the different performance of the two modes, the gripper crushing the tofu in the stiff mode while gently picking it up in the soft mode is presented in Fig. 7c. The whole process is facilitated by only one actuator as shown in Supplementary Video 5. The comparison of this quadra-stable gripper with the bi/multi-stable grippers in the literature is provided in Fig. 7d, where the gripper in this paper is the only one that can achieve bistability and variable stiffness with 1 actuator. This proves the effectiveness of our method in designing easy-to-actuate multi-stable structures. In the future, the shape and compliance of the end effectors can be designed carefully for better object holding, and the scaling-down for medical endoscope devices can be carried out.*

Figure 7. Variable stiffness quadra-stable gripper. **a** Demonstration of a gripper with both stiff mode and soft mode based on a quadra-stable structure. Two of stable states are the open and closed state of the gripper in stiff mode, where the energy barrier during the reconfiguration is much higher than that between the other two of stable states in soft mode. **b** The grip force data of the gripper by the experiment. With different opening distances between the gripper's end effectors, the variations of the gripping force, i.e. a constant force, in the stiff mode and soft mode are denoted by the orange curve and the blue curves. The gripping force in the stiff mode is around 30 times of the force in the soft mode. **c** The performance that the gripper grabbed a tofu in different modes. The gripper crushed the tofu in the stiff mode while picked up the tofu intactly in the soft mode (see Supplementary Video 5 for the whole process.) **d** Different types of bi/multi-stable grippers in the literature that are compared in terms of the number of needed actuators, the number of functional modes, the number of distractive stable states and part counts.

Comment 1: In Figure 1, there is a red star to represent the authors' work. Highlighting or using some method to make it more clear where this work fits within the context of others would be helpful.

Response: Thank you for the insightful comment. To address this concern, we added the description to Figure 1a as shown in bold font below.

Figure 1a: Literature survey of different types of the multi-stable structures in terms of their number of designable stable states and the number of actuators needed for the reconfiguration. **Heavy lines in different colors bound the upper limits of performance of the corresponding researches.**

To strengthen the relationship between our work and others further, we added a radar graph to illustrate the comparison of this quadra-stable gripper with the bi/multi-stable grippers, where the gripper in this paper is the only one that can achieve bistability and variable stiffness with 1 actuator.

Comment 2: In Figure 1 on the x-axis, there is "1D", "2D", or "3D" below each of the designable states. It is unclear what this refers to.

Response: Thank you for pointing out this confusion. The explanation is added on the x-axis as shown in **Fig. 1a**.

Reviewer #4

General Comment: Multi-stable structures are especially attractive for the application of morphing because they can be switched from one stable configuration to another with little energy and can be locked in any of the stable configurations without the need for constant actuation. However, most multi-stable structures consist of bistable unit cells and have undesirable stable configurations, which pose challenges in transforming to required states using easy-to-implement actuation techniques. The authors proposed multi-compatible truss structures with limited redundant stable states, obtained the minimum energy paths for switching between required states, and demonstrated the concept without the need for accurate actuation for configuration-switching. Note that although the topic is very interesting, it is technically too specific for a general journal and most importantly, the adopted design method is well developed, and the quality of the paper does not meet the high standard of Nature Communications. The paper is thus not recommended for publication in Nature Communications.

Response: Thank you for the comments and for the time and dedication in reviewing our manuscript. We agree that the numerical methods adopted in this paper, such as the constrained optimization tool and the Nudged Elastic Band (NEB) method, have no novelty indeed. The novelty of this paper is about (1) presenting a previously unknown method for constructing multi-stable structures, which is through employing the newly proposed multi-compatibility constraints, and (2) providing a following paths analysis and inverse actuation design to achieve “easy actuation,” i.e., facilitating specified reconfiguration with only one inaccurate actuator. The numerical methods employed are all well-established. To demonstrate the novelty and usefulness of this design method, we added an application of a gripper with varying stiffness in Fig. 7 and Supplementary Video 5. The following paragraphs were added to Page 13.

Page 13 Line 288: To explore the possible application of the proposed easy-to-actuate multi-stable structures, a variable stiffness gripper is designed employing a quadra-stable structure (Fig. 7a). It can reconfigure between 4 stable states with only 1 actuator. The reconfiguration between 2 (out of 4) stable states gives the stiff gripping mode, while the other 2 stable states provide the soft mode. One actuator is enough to operate the gripper in the two respective modes as well as switch between the two modes. Fig. 7b presents the specific force data of the gripper, where the gripping force in the stiff mode is around 30 times of the force in the soft mode. The measurement setup is presented in Supplementary Information, Section 9. To demonstrate the different performance of the two modes, the gripper crushing the tofu in the stiff mode while gently picking it up in the soft mode is presented in Fig. 7c. The whole process is facilitated by only one actuator as shown in Supplementary Video 5. The comparison of this quadra-stable gripper with the bi/multi-stable grippers in the literature is provided in Fig. 7d, where the gripper in this paper is the only one that can achieve bistability and variable stiffness with 1 actuator. This proves the effectiveness of our method in designing easy-to-actuate multi-stable structures. In the future, the shape and compliance of the end effectors can be designed carefully for better object holding, and the scaling-down for medical endoscope devices can be carried out.

Figure 7. **Variable stiffness quadra-stable gripper.** **a** Demonstration of a gripper with both stiff mode and soft mode based on a quadra-stable structure. Two of stable states are the open and closed state of the gripper in stiff mode, where the energy barrier during the reconfiguration is much higher than that between the other two of stable states in soft mode. **b** The grip force data of the gripper by the experiment. With different opening distances between the gripper's end effectors, the variations of the gripping force, i.e. a constant force, in the stiff mode and soft mode are denoted by the orange curve and the blue curves. The gripping force in the stiff mode is around 30 times of the force in the soft mode. **c** The performance that the gripper grabbed a tofu in different modes. The gripper crushed the tofu in the stiff mode while picked up the tofu intactly in the soft mode (see Supplementary Video 5 for the whole process.) **d** Different types of bi/multi-stable grippers in the literature that are compared in terms of the number of needed actuators, the number of functional modes, the number of distractive stable states and part counts.

Comment 1: The authors presented a study of easy-to-actuate multi-compatible truss structures with prescribed reconfiguration. They adopted a two-step method, i.e., structure optimization followed by optimizing actuation. A better strategy would be formulating a single optimization problem with both prescribed reconfigurations and actuation as constraints.

Response: Thank you for the valuable comment. There are three sequential steps instead of two: (1) multi-compatible structural design, satisfying multi-compatibility constraints, and minimizing the cost function in terms of bars' lengths, (2) MEP finding, minimizing two parts of energy while satisfying the orthogonality requirement of

the gradient in an iteration solving scheme, and (3) actuation inverse design, satisfying actuator's lengths decreasing monotonically in constraints, and maximize the cost function in term of the variation of the actuator length. The three steps are decoupled. First, we solve the structural geometry based on multi-compatibility constraints. Then, based on the designed geometry of the multi-compatible structure, we explored the minimum energy paths between stable states. Finally, based on the calculated reconfiguration paths, we designed actuation following the paths.

The three steps are not merged in this paper. This is because it is enough for this paper to propose multi-compatibility constraints and inversely design actuators following the MEP to design easy-to-actuate multi-stable structures and it is not essential to merge three steps. However, there are benefits for merging as this reviewer correctly suggested, which enables us to optimize the multi-compatible structure together with the actuation to obtain a better actuation design, such as minimizing the distance between the actuator mounting position and the joints. It is meaningful to refine the design procedure in the future, and we preliminarily attempted to construct a loop algorithm with the warm-start optimization for connecting the three steps, as shown in Supplementary Fig. 7a in SI Section 5.1, and added the following sentences to Page 6.

Page 6 Line 159: *The design procedure involves a three-step optimization: 1) compute the multi-compatible structure geometry and stiffness; 2) calculate the MEP by NEB method; 3) design the corresponding actuation. A higher-level loop has been applied on top of the three-step optimization for a more compact actuation design and energy barrier control. The detailed algorithms are provided in Supplementary Information, Section 5. The three optimization problems can be merged into one in the future for better computational efficiency.*

Page 6 Line 166: *In this paper, we designed the multi-compatible structures and actuators in sequence.*

SI Section 5: *The multi-compatible structure can be designed with many other constraints or targets, such as tailoring the energy barrier or together with the actuation design for a more suitable actuator. However, achieving these further targets requires information on the MEP calculated by the NEB method, an iteration procedure. Therefore, we constructed a loop algorithm with the warm-start optimization, where the results obtained from a previous optimization are used as the initial guess or starting point for a subsequent optimization process. Generate a preliminary multi-stable structure by the first optimization Opt.1 and employ the NEB method to obtain the MEP. Then input this preliminary designed structure as the initial guess to the second optimization Opt.2, whose constraints based on the calculated MEP are added to the multi-compatibility constraints. The loop algorithms for the entire easy-to-actuate multi-stable structure design and energy barrier tailoring are shown in Supplementary Fig. 7. [6]*

[6] Jingyi Yang et al. "Folding and deploying identical thick panels with spring-loaded hinges". In: *Extreme Mechanics Letters* 52 (2022), p. 101637.

Supplementary Fig. 7. **The flow chart of the warm-start optimization, where the results obtained from a previous optimization are used as the initial guess or starting point for a subsequent optimization process. a** The warm-start optimization for the actuation design. **b** The warm-start optimization for tailoring energy barriers.

SI Section 5.1: Algorithm for combining the structure design and actuation design

In Supplementary Fig. 7a: generate a multi-stable structure $G1$ that can achieve prescribed configurations in Opt.1 and calculate its reconfiguration path MEP 1 through the NEB method. Input the structure design $G1$ as the initial guess of another optimization Opt.2, which satisfies the multi-compatibility constraints and the actuation design constraints based on MEP 1. Then perform a kinematic simulation on the results of Opt.2, and check whether the calculated actuator can actuate the designed multi-stable structure.

Supplementary Fig. 8 demonstrates design results $G2$, AC1 of Opt.2 withstanding the verification of the kinematic simulation. The red, green and blue areas denote three prescribed configurations of the moving plate. The red dashed lines denote the designed actuator, whose length decreases monotonically. The black dashed lines mean the endpoint of the calculated actuator is rigidly connected to the moving plate. Furthermore, the same optimization Opt.3 as Opt.2 can be performed more times on $G2$, AC1 until a more high-standard condition for the actuation converges, such as the ratio of the distance between the actuator mounting position on the moving plate and the reference point to the maximum geometric size of the moving plate.

Supplementary Fig. 8. **The result of the double-optimization for the entire design procedure, i.e. combining the structure design and actuation design into one-step constrained optimization.** The red dashed lines denote the designed actuator, whose length decreases monotonically. The black dashed lines mean the endpoint of the calculated actuator is rigidly connected to the moving plate.

Comment 2: The authors presented an inverse design method for multi-compatible truss structures. Two examples are given. The first is a two-plate-four-bar structure (2P4B), and the second is a three-plate-six-bar (3P6B) structure. Although these two examples are already relatively simple, self-contact between components emerges. The component interference is expected to increase dramatically with the complexity of the structure. However, it seems that the authors could only cope with the interference in an ad hoc manner. In fact, they were only able to show the experimental results for the simpler example (i.e., 2P4B). Without taking the interference into account in the design stage, the obtained results by the method may not be as useful as one expects if much more complex structures are preferred.

Response: Thank you for pointing out the confusion. This paper also demonstrates the experimental results for the more complex example, i.e., 3P6B in Fig. 2d and Supplementary Video 2, and the layering analysis for this 3P6B structure is shown in Supplementary Information, Section 6.

We added interference-free constraints for simple structures in the design stage, which enables us to 3D print the structure integrally without the need for layering and assembly, as shown in Fig. 2e-g and SI Section 6.1. And we summarized the layering assignment rules for general structure layering instead of an ad hoc manner when layering is ineluctable, such as a crank in the structure, i.e. the crank bar rotating 360 degrees, as shown in SI Section 6.2. Finally, we took 3P6B structure as an example to illustrate the layering process according to the rules in SI Section 6.3.

Page 7 Line 204: We attempted to propose the interference-free constraints (see Supplementary Information, Section 6.1) so that we can design an interference-free two-plate-three-bar example as shown in Fig. 2e, where all components could be arranged in one plane after reshaping two plates.

Figure 2. **Examples of the multi-compatibility design.** **a** A two-plate-four-bar tri-stable structure (2P4B). The grey plate is fixed to the ground, the moving plate can achieve three prescribed configurations, denoted by three respective colors (green, blue, and purple). **b** The physical model corresponding to **a**, where the structure is multi-compatible at three prescribed configurations without component deformation, while the white bars made of TPU deform significantly during the reconfiguration. **c** A three-plate-six-bar tri-stable structure (3P6B). The grey plate is fixed to the ground, the other two moving plates can achieve three prescribed configurations, denoted by three colors (green, blue, and purple). **d** The physical model corresponding to **c** where the components show significant deformation during reconfiguration. **e** A design of a real-2D two-plate-three-bar tri-stable structure, where all components could be arranged in one plane after reshaping, is obtained by adding the interference-free constraints. **f** The physical model corresponding to **e**. **g** A sequential assembly of four units in **f**, which can be stable with three different curvatures.

SI Section 6.1 (interference-free constraints for simple 2P4B structure): Interference-free constraints ensure any two bars located on the same layer should be interference-free during kinematical motion. If there is a crank in the structure, i.e. one of the bars rotates 360 degrees, it would be inevitable to layer. Therefore, the limitations of the components' motions are defined firstly:

$$\max \text{ang}(\mathbf{B}_i^b \mathbf{G}_i^b, \mathbf{B}_j^b \mathbf{G}_j^b) < \pi, i, j = 1, \dots, nc, i \neq j, b = 1, \dots, nb \quad (44)$$

where nc and nb are the total numbers of configurations and bars, i and j denote the serial number of the configuration, b denotes the serial number of the bar.

Allowing for designing the profiles of links rather than using straight links connecting joints directly, multi-stable structures achieving the prescribed configurations would be designed in a much larger design space. Inspired by the reference [6], the lengths of bars have constraints as shown in Supplementary Fig. 10:

$$|\mathbf{B}_1^b \mathbf{G}_1^b| < \min\{|\mathbf{G}_1^b \mathbf{G}_1^a|, |\mathbf{G}_1^b \mathbf{G}_1^c|\} \quad (45)$$

where a and c denote the adjacent joints to the b th joint. In Supplementary Fig. 10, red shaded areas are the swept areas of the red bars, the grey segments in between the grounded pivot points denote the available space for reshaping the profiles of the black links.

Supplementary Fig. 10. **The illustration of the interference-free constraints.** The red areas denote the swept areas of red bars, the grey segments in between the grounded pivot points denote the available space for reshaping the profiles of the black links.

SI Section 6.2 (layering assignment method): Assuming all of the bars are located on the same layer, the interference of components during the transformation is analyzed through the path calculated through the NEB method. Specifically, set one component as the reference system and plot the motion of the other components with respect to this component until all components have been analyzed, which seems we stand on this component and observe the motions of the other components. Then in accordance with the above collision analysis, the components could be assigned to different layers following the below rules:

Rule 1: two adjacent links that are paired by the same joint should be assigned to different layers. [6]

Rule 2: two bars that collide with each other should be assigned to different layers.

Rule 3: if i th bar collides with k th joint, the i th bar should be assigned to the outermost layer (top or bottom) of all the bars connected by the k th joint. [7]

[6] Hiroaki F. and Kiyoshi O. On the design of planar mechanisms with consideration of interferences of moving links. *Bulletin of JSME* 27, 341–347 (1984).

[7] Ran Z., Thomas A., and Bernd B. Computational design of planar multistable compliant structures. *ACM Transactions on Graphics (TOG)* 40, 1–16 (2021).

Comment 3: The authors adopted the nudged elastic band method (NEB) to obtain the minimum energy reconfiguration path for the multi-compatible truss structures. Note that this method is well-developed in computational chemistry. Details of the method, e.g., Eqs. (3–6) and the associated texts, can be moved from the main text to Supplementary Information.

Response: Thank you for the suggestion. We have moved the introduction of the NEB method to Supplementary Information and leave the citation of Supplementary Information. The revised sentences are added to the position corresponding to Page 4.

Page 4 Line 143:, where $C1$ and CN represent the configuration points of the two respective stable states that are designed (see Supplementary Information, Section 4.2 for the NEB method in detail.)

REVIEWERS' COMMENTS

Reviewer #1 (Remarks to the Author):

The revised version of the manuscript "Easy-to-actuate multi-compatible truss structures with prescribed reconfiguration" has effectively addressed all of my previous comments. I particularly appreciate the attention given to ensuring the reproducibility of the results, with the addition of detailed algorithms and a code that I have verified to be functional. With these improvements, I believe the manuscript is now suitable for publication in its current form.

Reviewer #1 (Remarks on code availability):

The provided code is functional and adds value to the paper. However, I suggest improving the comments within the code to enhance readability. While I found it slightly challenging to read, I did not identify any significant issues with the code.

Reviewer #2 (Remarks to the Author):

I am grateful to the author for their careful responses to my previous questions. From my point of view, the current version of the work can be published.

Reviewer #3 (Remarks to the Author):

The authors have done a nice job responding to the comments.

Reviewer #4 (Remarks to the Author):

I am satisfied with the revisions made by the authors. The paper is now recommended for publication in Nature Communications.

Reviewer #5 (Remarks to the Author):
